# Deciphering the contributing motifs of reconstructed cobalt (II) sulfides catalysts in Li-CO$_2$ batteries

Yingqi Liu[1,7], Zhiyuan Zhang[1,7], Junyang Tan[1], Biao Chen [2], Bingyi Lu[1], Rui Mao[1], Bilu Liu [1], Dashuai Wang [3] ✉, Guangmin Zhou [1] ✉ & Hui-Ming Cheng [4,5,6] ✉

Developing highly efficient catalysts is significant for Li-CO$_2$ batteries. However, understanding the exact structure of catalysts during battery operation remains a challenge, which hampers knowledge-driven optimization. Here we use X-ray absorption spectroscopy to probe the reconstruction of CoS$_x$ (x = 8/9, 1.097, and 2) pre-catalysts and identify the local geometric ligand environment of cobalt during cycling in the Li-CO$_2$ batteries. We find that different oxidized states after reconstruction are decisive to battery performance. Specifically, complete oxidation on CoS$_{1.097}$ and Co$_9$S$_8$ leads to electrochemical performance deterioration, while oxidation on CoS$_2$ terminates with Co-S$_4$-O$_2$ motifs, leading to improved activity. Density functional theory calculations show that partial oxidation contributes to charge redistributions on cobalt and thus facilitates the catalytic ability. Together, the spectroscopic and electrochemical results provide valuable insight into the structural evolution during cycling and the structure-activity relationship in the electrocatalyst study of Li-CO$_2$ batteries.

The overreliance on fossil fuels has significantly increased atmospheric CO$_2$ levels, which poses a severe threat to the environment and the survival of humankind[1]. To mitigate global warming and climate change, it is vital to develop carbon-neutral technologies, including CO$_2$-related technology, clean energy technology, as well as high energy-density energy storage systems[2-4]. Recently, Li-CO$_2$ batteries have emerged as an attractive solution due to their dual functions of energy storage capability and CO$_2$ recyclability[5-9]. Based on the reaction 4Li + 3CO$_2$ ↔ 2Li$_2$CO$_3$ + C, Li-CO$_2$ batteries have a high theoretical potential of 2.8 V vs Li/Li$^+$ and a theoretical energy density of 1876 Wh kg$^{-1}$. However, the sluggish kinetics of CO$_2$ reactions cause unsatisfactory electrochemical performance, such as high overpotential, poor reversibility, low energy efficiency, etc. Therefore, there is a critical need to develop highly efficient catalysts that can unlock the full potential of this emerging technology[3,10-13].

Transition metal sulfides show superior catalytic abilities in Li-CO$_2$ batteries as shown in Fig. 1a and Supplementary Table 1[10,14-41]. However, these sulfide catalysts are susceptible to irreversible reconstruction, particularly oxidation, due to their thermodynamic instability. In Li-CO$_2$ batteries, the main discharge product is Li$_2$CO$_3$, whose decomposition may generate singlet O$_2$ or superoxide radicals during charging[9,42-45]. The formation of these aggressive oxygen species can exacerbate oxidation phenomena, influencing the activity of sulfides based on their reconstructed structures and oxidized states. Complete

[1]Tsinghua-Berkeley Shenzhen Institute & Tsinghua Shenzhen International Graduate School, Tsinghua University, Shenzhen 518055, PR China. [2]School of Materials Science and Engineering and Tianjin Key Laboratory of Composite and Functional Materials, Tianjin University, Tianjin 300350, PR China. [3]Institute of Zhejiang University-Quzhou & Key Laboratory of Biomass Chemical Engineering of Ministry of Education, College of Chemical and Biological Engineering, Zhejiang University, Hangzhou 310027, China. [4]Shenyang National Laboratory for Materials Science, Institute of Metal Research, Chinese Academy of Sciences, Shenyang 110016, China. [5]Institute of Technology for Carbon Neutrality, Shenzhen Institute of Advanced Technology, Chinese Academy of Sciences, Shenzhen 518055, PR China. [6]Shenzhen University of Advanced Technology, Shenzhen 518055, China. [7]These authors contributed equally: Yingqi Liu, Zhiyuan Zhang. ✉e-mail: dswang@zju.edu.cn; guangminzhou@sz.tsinghua.edu.cn; hm.cheng@siat.ac.cn; cheng@imr.ac.cn

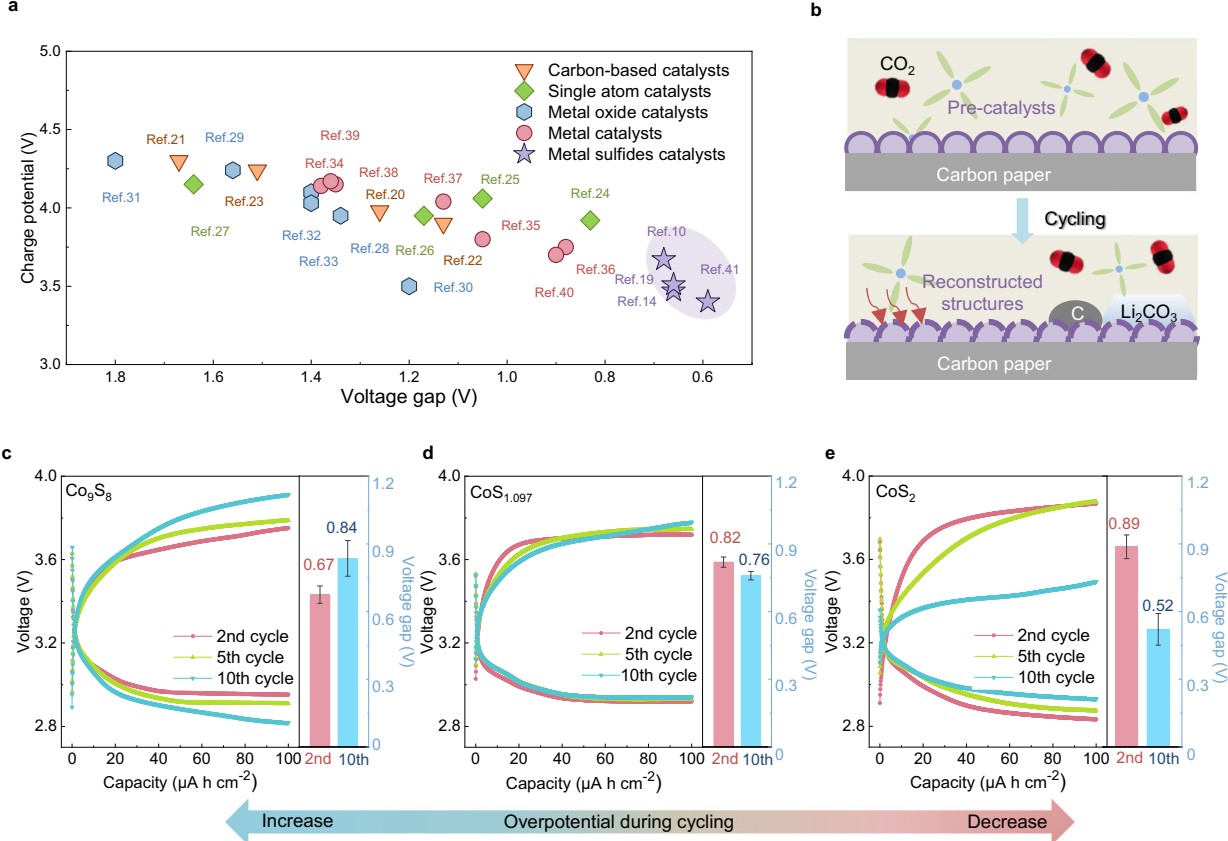

**Fig. 1 | Electrochemical behaviors of CoSx during cycling. a** Performance comparison of reported catalysts. (Carbon-based catalysts reported in refs. 20–23.; Single-atom catalysts reported in refs. 24–27.; Metal oxide catalysts reported in refs. 28–33; Metal catalysts reported in ref. 34–40; Metal sulfide catalysts reported in refs. 10,14,19,41) **b** Schematic of catalyst reconstruction in Li-CO₂ batteries. Discharge and charge curves of **c** Co₉S₈, **d** CoS₁.₀₉₇, and **e** CoS₂ with a limited capacity of 100 μA h cm⁻² at a rate of 20 μA cm⁻². Error bars of the voltage gap represent the standard deviation from three independent measurements. Source data are provided as a Source Data file.

oxidation typically induces structural changes and widens the band gap, similar to oxides, thereby significantly reducing activity. Conversely, oxysulfides sometimes exhibit higher stability and activity compared to sulfides, making them more appealing[46,47]. Therefore, it is crucial to consider sulfides as pre-catalysts, investigate their structural reconstruction, assess the impact of oxygen during cycling, and identify the actual active structures (Fig. 1b). This will help understand active motifs and intrinsic properties in structural adaptation under battery operation, enabling the development of advanced catalysts for Li-CO₂ batteries. Nevertheless, little attention has been given to it so far.

To address the critical aspects mentioned above, we investigate the electrochemical performance and structural evolution of three types of cobalt (II) sulfide pre-catalysts (CoSₓ, x = 8/9, 1.097, and 2) in Li-CO₂ batteries. We find that the CoS₂ cathode has a reduced overpotential, while CoS₁.₀₉₇ and Co₉S₈ do not show the same decrease during cycling. The spectroscopic analysis indicates that the oxidation of the CoS₂ cathode terminates with Co-S₄-O₂ motifs while CoS₁.₀₉₇ and Co₉S₈ are completely oxidized with a structure similar to CoO in Li-CO₂ batteries. Supported by titration results, we propose that the oxidation states after reconstruction are affected by side reactions during charge on the pre-catalyst. Density functional theory (DFT) calculations revealed that partial oxygen substitution modulates the electronic structure and shifts the d-band center to higher energy, thus improving the catalytic ability of CoS₂. Hence, CoS₂ has a high performance with an overpotential of 0.43 V after 400 h, while the overpotentials of CoS₁.₀₉₇ and Co₉S₈ cathodes exceed 2 V after only 200 h in Li-CO₂ batteries. This work provides insights into catalyst reconstruction

under complex environments and contributes significantly to understanding the inherent structure-activity relationship in Li-CO₂ batteries.

## Results and discussion
### Structure characterizations and electrochemical behaviors of CoSₓ

The CoSₓ (x = 8/9, 1.097, and 2) samples were synthesized by sulfidation of Co(OH)₂ nanosheet arrays electrodeposited on pieces of carbon papers (CP) (Supplementary Fig. 1). Their X-ray diffraction (XRD) patterns in Supplementary Fig. 2 contain diffraction peaks that match those of standard CoS₂, CoS₁.₀₉₇, and Co₉S₈ except for peaks at about 26°, 43° and 55°(2θ), which correspond to the CP substrate[7,14,48]. The morphologies of the as-prepared CoSₓ were studied using scanning electron microscopy (SEM). Supplementary Fig. 3a–d show that CoS₂, CoS₁.₀₉₇, and Co₉S₈ maintain a similar nanosheet structure to electrodeposited Co(OH)₂. Supplementary Fig. 4 also confirms that the electrochemical surface active area (ECSA) of the three cathodes is similar, ruling out their influence on the following electrochemical test. The high-resolution transmission electron microscopy (HRTEM) images in Supplementary Fig. 5a–c show three sets of lattice fringes, with interplanar spacings of 0.28 nm, 0.29 nm, and 0.30 nm, which can be assigned to (200), (204), and (311) planes of CoS₂, CoS₁.₀₉₇, and Co₉S₈, respectively. Element mappings in Supplementary Fig. 6a–c confirm the uniform distribution of Co and S. Their electrochemical behaviors exhibit notable differences in Li-CO₂ batteries, as shown in Fig. 1c–e and Supplementary Fig. 7. Batteries with Co₉S₈ cathodes show an increase in overpotential, while the discharge and charge curves for

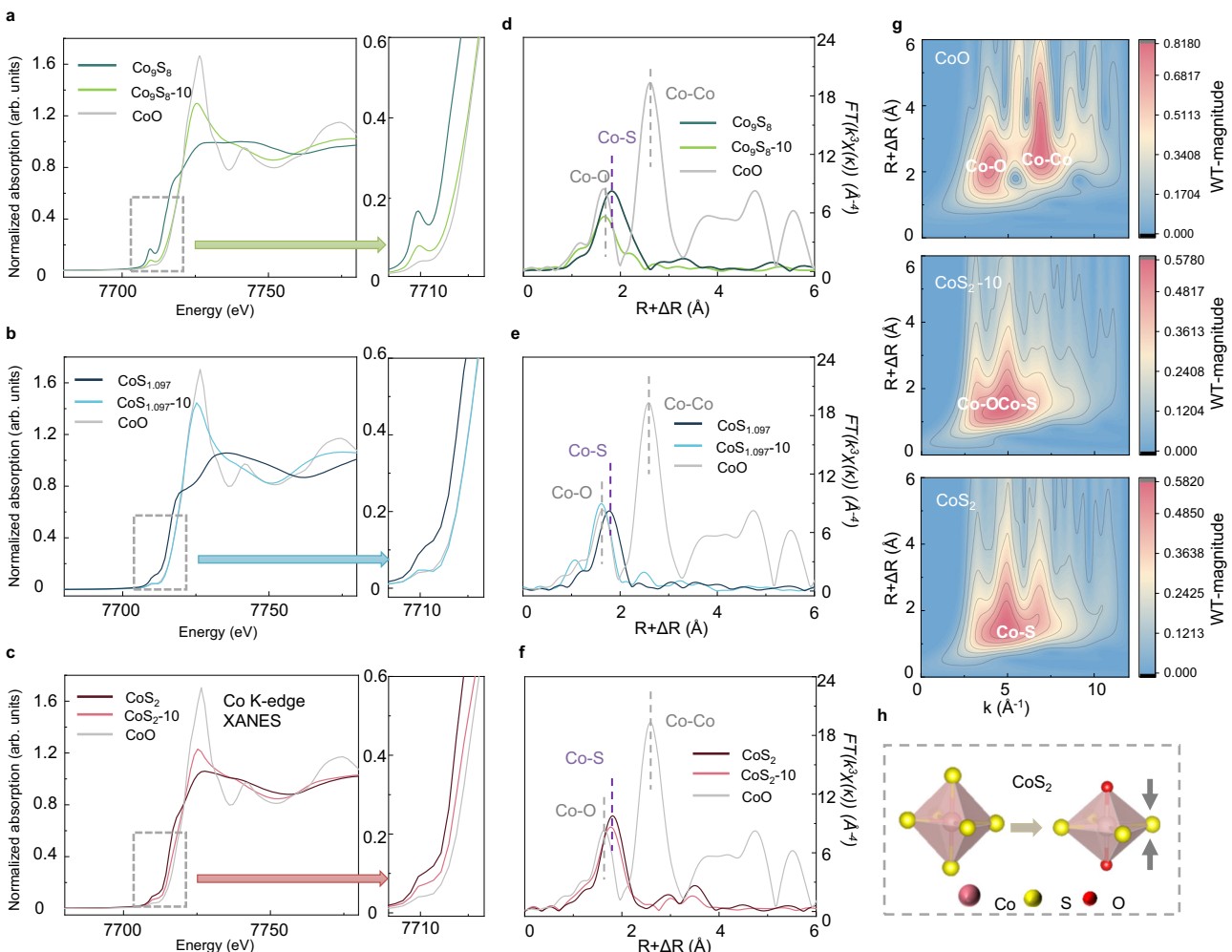

**Fig. 2 | The structural evolution characterizations.** Co *K*-edge XANES of **a** Co₉S₈, **b** CoS₁.₀₉₇, and **c** CoS₂ before and after 10 cycles. Co *K*-edge FT-EXAFS of **d** Co₉S₈, **e** CoS₁.₀₉₇, and **f** CoS₂ before and after 10 cycles. **g** WT-EXAFS of CoO, CoS₂-10 and CoS₂. **h** Schematic of structural evolution of CoS₂ during cycling. Source data are provided as a Source Data file.

the CoS₁.₀₉₇ cathodes only have a slight change in 10 cycles. Interestingly, the charge plateau of CoS₂ for 10 cycles is much lower than the pristine one, and the overpotential is reduced from 0.89 V to 0.52 V. These differences prompt our investigation into the $CO_2$ reaction and real active structure within Li-$CO_2$ batteries.

## The structural evolution characterizations

To investigate structural evolutions that affect the electrochemical behaviors, we performed X-ray photoelectron spectroscopy (XPS) and X-ray absorption spectroscopy (XAS) analyses on the cathodes at different cycles. The XPS results in Supplementary Figs. 8 and 9 indicate that S in CoS₂ has less than the full coordination, which maintains its structure during cycling in Li-$CO_2$ batteries. In contrast, Co-S bindings decrease significantly in both CoS₁.₀₉₇ and Co₉S₈, suggesting severe structural changes during cycling. To further confirm the specific structure after cycling, we performed XAS at Co *K*-edge to study the evolution of Co₉S₈, CoS₁.₀₉₇, and CoS₂ before and after 10 cycles (labeled as CoSₓ-10) in Fig. 2a–c. X-ray absorption near-edge structure (XANES) in Fig. 2a, b show that the absorption edges of Co₉S₈ and CoS₁.₀₉₇ shift to higher energy and overlap with that of CoO after cycling, indicating sulfide oxidation. Fourier transformed extended X-ray absorption fine structures (FT-EXAFS) in Fig. 2d, e show that the first coordination shell of Co₉S₈-10 and CoS₁.₀₉₇-10 is much shorter than that of the pristine samples and are close to that of CoO, confirming most S atoms being substituted by O atoms. In contrast, for

CoS₂, the absorption edge of CoS₂-10 is located between that of pristine CoS₂ and CoO, and the first coordination shell of Co is contracted but still longer than that of Co-O (Fig. 2c, f). We compare the FT-EXAFS of CoS₂-10 with FEFF[49]-calculated Co-O path and Co-S path, showing that the oscillation in the first coordination shell of CoS₂ can not be solely assigned to Co-O scattering or Co-S scattering (Supplementary Fig. 10 a–c). The element of scattering atoms can be derived from EXAFS by the energy dependence of ossilcation amplitude[50]. Therefore, we performed inverse Fourier transformation to study the coordination atoms, showing that the first coordination shell is composed of Co-S and Co-O (Supplementary Fig. 10 d–f). Wavelet-transformed EXAFS (WT-EXAFS) shows that the maximum in the region of the first coordination shell is overlapped by Co-O scattering and Co-S scattering, further confirming that Co is coordinated by both S and O in CoS₂-10 (Fig. 2g). The local structure of Co is quantitatively studied by the least-squares fitting of EXAFS, showing that Co atoms are coordinated by four S atoms at 2.27 Å and two O atoms at 2.00 Å (Supplementary Fig. 11 and Supplementary Table 2). Therefore, we speculate that the CoS₂ is reconstructed to cobalt-oxysulfide, as shown in Fig. 2h.

## $CO_2$ reduction and evolution reaction mechanism

The cyclic voltammetry (CV) curves of Co₉S₈, CoS₁.₀₉₇, CoS₂, and CP in $CO_2$ and Ar atmosphere are shown in Supplementary Fig. 12. All batteries exhibit featureless curves in the Ar atmosphere while obvious oxidation and reduction peaks in the $CO_2$ atmosphere, which indicates

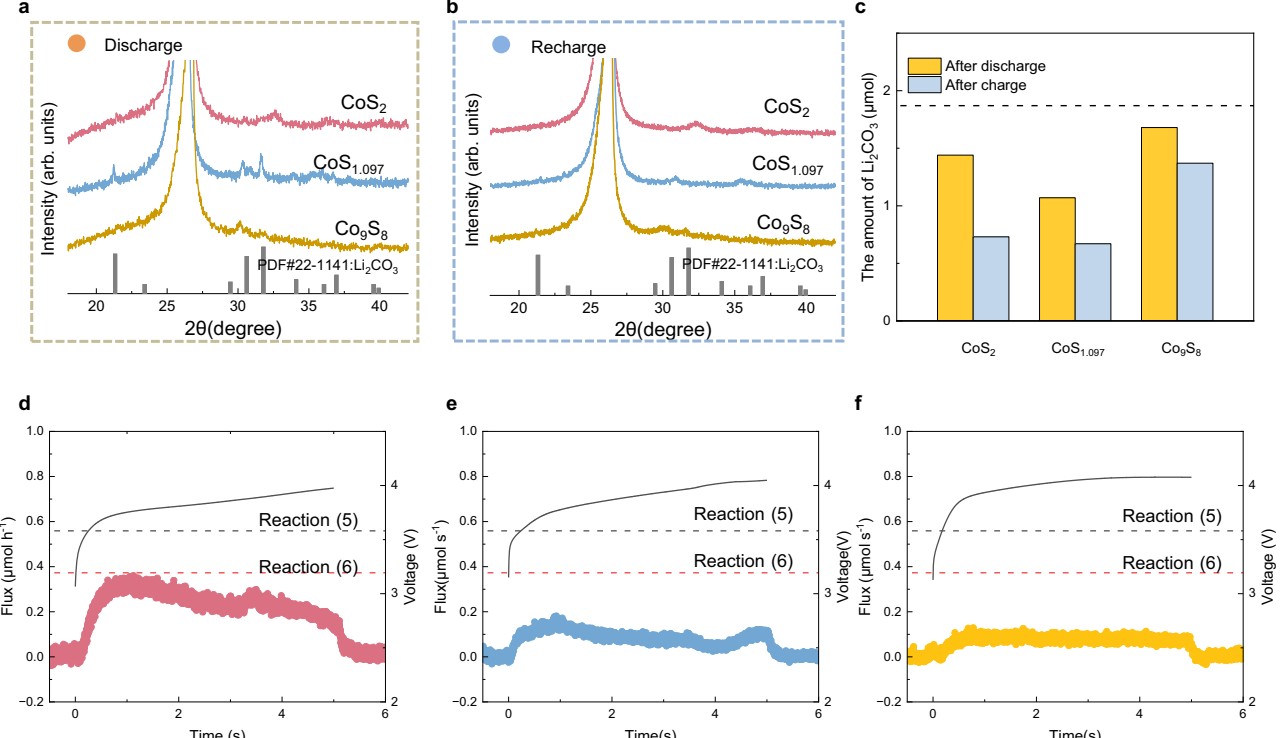

**Fig. 3 | $CO_2$ reduction and evolution reaction mechanism.** XRD patterns of the three cathodes for **a** discharging and **b** charging to 200 µA h $cm^{-2}$. **c** The amount of $Li_2CO_3$ formation and residues on three catalysts after discharge and charge. The dashed line is the theoretical value of $Li_2CO_3$ formation after discharge. DEMS results of **d** $CoS_2$, **e** $CoS_{1.097}$, and **f** $Co_9S_8$ during charge at a current density of 20 µA $cm^{-2}$ with a limited capacity of 100 µA h $cm^{-2}$. The dashed lines are the expected flux charge reactions (5) and (6) based on the applied current density. Source data are provided as a Source Data file.

the electrochemical inactivity of sulfides and substrate without $CO_2$ at the range of 2.2-4.7 V. Therefore, the reactions during discharge and charge mainly rely on $CO_2$ for the cathodes. To gain mechanistic insight into the electrocatalytic process, ex situ characterizations of products on $CoS_x$ electrodes after discharge and charge were first performed, as shown in Supplementary Fig. 13, including SEM (Supplementary Fig. 14), XRD (Fig. 3a, b), Raman spectroscopy (Supplementary Fig. 15). The SEM images show that the discharge products are large and rodlike covering the surface of $CoS_{1.097}$, while those on $CoS_2$ and $Co_9S_8$ are smaller in Supplementary Fig. 14a–c. The XRD patterns in Fig. 3a show the signal of discharge products can be assigned to $Li_2CO_3$ (#PDF22-1141) on $CoS_{1.097}$. After the charge, even though no other peaks are on all cathodes in Fig. 3b, the irregular residues can be easily observed on $CoS_{1.097}$ and $Co_9S_8$ while those on $CoS_2$ are not observable in Supplementary Fig. 14d–f. These results roughly indicate that, in comparison with $CoS_{1.097}$ and $Co_9S_8$, $CoS_2$ has a higher reversibility. As the discharge products on $CoS_2$ and $Co_9S_8$ can not be clearly identified, we infer $Li_2CO_3$ most probably is the discharge product for the three sulfides based on previous reports and XRD pattern of discharged $CoS_{1.097}$ in Li-$CO_2$ batteries[12,51,52]. We also performed Raman spectroscopy in Supplementary Fig. 15 but peaks at 1080 $cm^{-1}$ corresponding to vibration of $Li_2CO_3$ are weak on discharged $CoS_2$ and $Co_9S_8$.

To verify our assumption and quantify the reversibility for the three catalysts in Li-$CO_2$ batteries, titration experiments by phosphoric acid are performed on the catalysts after discharge and charge under a current density of 20 µA $cm^{-2}$ with a limited capacity of 100 µA h $cm^{-2}$, which consistent with electrochemical test (Supplementary Fig. 16)[53]. As shown in Supplementary Fig. 17, $CO_2$ generation after titrating acid solution on the discharged catalyst, suggesting carbonates, most likely $Li_2CO_3$ based on the above results, are discharge products on the three catalysts. With external standard 1# in Supplementary Fig. 18 and

Supplementary Table 3, the quantities of formed and residual $Li_2CO_3$ during discharge and charge on the three cathodes are shown in Fig. 3c and Supplementary Table 4. By now, the reported possible discharge reactions in Li-$CO_2$ batteries are shown as following reactions (1)–(4)[9,54–56]. The charge to mass of $Li_2CO_3$ in all reactions is $2e^-$/$Li_2CO_3$, including reaction (2) if $Li_2C_2O_4$ disproportionates to $Li_2CO_3$. For a $2e^-$/$Li_2CO_3$ process, ~57–89% of the discharge process goes to the formation of $Li_2CO_3$, indicating $Li_2CO_3$-related reactions are dominant during discharge for the three sulfides. For charge, there are two possible reactions (5) and (6) with $Li_2CO_3$ decomposition[9,43].

$$4Li^+ + 4e^- + 3CO_2 \rightarrow 2Li_2CO_3 + C \tag{1}$$

$$4Li^+ + 4e^- + 2CO_2 \rightarrow 2Li_2C_2O_4 \tag{2}$$

$$2Li^+ + 2e^- + + 2CO_2 \rightarrow Li_2CO_3 + CO \tag{3}$$

$$2Li^+ + 2e^- + CO_2 + O \rightarrow Li_2CO_3 \tag{4}$$

$$2Li_2CO_3 + C \rightarrow 4Li^+ + 3CO_2 + 4e^- \tag{5}$$

$$Li_2CO_3 \rightarrow 2Li^+ + CO_2 + 1/2O_2/O + 2e^- \tag{6}$$

The ratio of $Li_2CO_3$ and $CO_2$ is ~0.67 for reaction (5) and 1 for reaction (6). In situ differential electrochemical mass spectrometry (DEMS) analysis was performed to calculate $CO_2$ generation during the charge on the three sulfides. Figure 3d–f and S19 show that only $CO_2$ (m/z = 44) generation can be observed on all cathodes and the amount of $CoS_2$ is much higher than that on $Co_9S_8$ and $CoS_{1.097}$. As the

numerical results are summarized in Supplementary Table 5, the ratio of $Li_2CO_3$ to $CO_2$ of $CoS_{1.097}$ is 0.85, close to 1, indicating that reaction (6) may mostly happen during charge. Even though no signal of $O_2$ (m/z = 32) has been observed, oxygen species generation is commonly possible and threatens the catalyst's durability[43]. Consequently, $CoS_{1.097}$ is oxidized to CoO during cycling and affects its electrochemical performance. The ratio of $Co_9S_8$ and $CoS_2$ is ~0.76 and ~0.56 respectively, close to that of the reaction (5). However, the conversion efficiency of $Li_2CO_3$ on $Co_9S_8$ is only 18.4% much lower than the other cathodes, indicating oxidation reactions mainly happened to supply capacity. The XAS results in Fig. 2 show that the valence states of Co in three sulfides maintain +II after 10 cycles, excluding the possibility of Co contribution to the charge capacity. Instead, the sulfur oxidation may be responsible for the charge capacity of $Co_9S_8$, as the decreased intensity of Co-S binding in Supplementary Fig. 8a. We also can't exclude the possibility of electrolyte decomposition that supplies the capacity and oxidizes the catalysts. By contrast, the higher charge efficiency of $CoS_2$ benefits its reconstruction to $Co-S_4-O_2$ instead of complete oxidation. We further titrated cathodes after the 9th charge, 10th discharge, and 10th charge to investigate the battery reaction on the reconstructed $CoS_2$. As shown in Supplementary Fig. 20, there was little $CO_2$ generation after titrating the cathodes after the 9th and 10th charge, while an obvious $CO_2$ generation peak can be observed on the cathode after the 10th discharge, suggesting that most $Li_2CO_3$ can be decomposed after charge during cycling. Since the measurement values of $CO_2$ generation on cathodes after the 9th and 10th charge are less than 5% of that on the cathode after the 10th discharge, we approximate the amount of $Li_2CO_3$ on the cathode after the 10th discharge as the quantities of $Li_2CO_3$ formation and decomposition in the 10th cycle. Based on external standard 2#, the amount of $Li_2CO_3$ formation is ~1.15 μmol, suggesting ~60% charge goes to form $Li_2CO_3$ during discharge in the 10th cycle (Supplementary Fig. 21 and Supplementary Table 3). These results demonstrate that $Li_2CO_3$ remains the main discharge product and can be almost completely decomposed during the charge on the reconstructed $CoS_2$ in cycling.

## DFT calculations and discussion

DFT calculations were performed to elucidate the relationship between sulfide structure and activity in $Li-CO_2$ batteries. Based on our experimental results, we constructed four substrates, including three pre-catalysts $Co_9S_8$, $CoS_{1.097}$, $CoS_2$ and oxygen partially substituted $CoS_2$ after cycling (denoted as $O-CoS_2$ in the latter discussion) in Supplementary Fig. 22. The adsorption energies of $CO_2$, Li, and $Li_2CO_3$ were first calculated to assess the interaction between substrates and reactants during charge and discharge in $Li-CO_2$ batteries as shown in Supplementary Figs. 23–26[18,57,58]. As summarized in Fig. 4a and Supplementary Table 6, the adsorption strengths of $CO_2$, Li, and $Li_2CO_3$ on $CoS_2$ are higher than those on other pre-catalysts, except that the adsorption energy of $CO_2$ is little weaker than that on $Co_9S_8$. Notably, a more negative value of adsorption energies on $O-CoS_2$ than $CoS_2$ signifies partial oxygen substitution effectively increases the adsorption strengths. The electronic modulation by O substitution is revealed in Supplementary Fig. 27 and Supplementary Table 7, showing the charge redistribution on neighboring cobalt atoms. A more positive region on $O-CoS_2$ than $CoS_2$ suggests that O substitution increases the local polarity and interaction with adsorbed species shown in Fig. 4b. Besides, the *d*-band center of Co shifts to a higher energy level related to the Fermi level due to oxygen substitution, also corresponding with increased adsorption strength of $O-CoS_2$ (Fig. 4c).

The Gibbs free energies at both open circuit (U = 0 V) and equilibrium (U = 2.85 V) potentials for five possible pathways on the four constructed catalysts to further determine the reaction kinetics are shown in Supplementary Figs. 28–32 and Tables S8–11. At the equilibrium potentials, Fig. 4d shows that *$CO_3$ and *C formation is the rate-determining step for three pre-catalysts and $CoS_2$ has the lowest energy difference of this step (2.46 eV) than $Co_9S_8$ (3.03 eV) and $CoS_{1.097}$ (3.02 eV). The rate-determining step of $O-CoS_2$ is changed to step (7) with ΔG (2.14 eV), also lower than that of three pre-catalysts, indicating that partial oxygen substitution further improves the catalytic ability of $CoS_2$ in $Li-CO_2$ batteries.

Based on our experimental evidence and DFT results, the structural evolution and consequent change in electrochemical performance are illustrated in Fig. 4d. $Co_9S_8$ and $CoS_{1.097}$ as pre-catalysts show serious parasitic reactions during charge in $Li-CO_2$ batteries. Consequently, the two catalysts have been fully oxidized during cycling, which passivates the catalytic abilities and results in increased overpotentials of batteries. On the contrary, $CoS_2$ with higher activity shows superior electrochemical performance and reversibility, of which oxidation is terminated and forms oxysulfide with $Co-S_4-O_2$ motif in $Li-CO_2$ batteries. The partial oxygen substitution increases the local polarity and the energy level of the d-band center, which adjusts the adsorption strength and thereby reduces the battery overpotential. In short, the initial properties of sulfides play a crucial role in their structural evolutions in batteries and thus affect the performance of batteries during cycling. Our finding also demonstrates the active motifs for reconstructed catalysts, which provide insights for understanding the high activity of sulfides and even other transition compounds.

## The application in $Li-CO_2$ batteries

The high activity of $CoS_2$ and increased energy efficiency owing to partial oxygen substitution are demonstrated in the latter electrochemical test. The CV curves in Supplementary Fig. 33 show a faster redox reaction rate between $CO_2$ and C on $CoS_2$, with oxidation and reduction currents being higher in comparison to $CoS_{1.097}$ and $Co_9S_8$. The onset potentials for $CO_2RR$ and $CO_2ER$ of three cathodes are compared corresponding to 0.25 mA cm$^{-2}$ shown in Fig. 5a[10,14]. $CoS_2$ exhibits more positive and lower potential for $CO_2RR$ (2.76 V) and $CO_2ER$ (4.27 V) in comparison to $CoS_{1.097}$ (2.69/4.39 V) and $Co_9S_8$ (2.59/4.42 V), indicating its higher catalytic activities. $CoS_2$ also shows high reversibility as higher charge capacity (1781.4 μA cm$^{-2}$) and Coulombic efficiency (CE) at 88.8% in the galvanostatic charge-discharge (GDC) test (Fig. 5b), while $CoS_{1.097}$, $Co_9S_8$ and CP have charge capacities of 1370.6, 940.0 and 47.7 μA cm$^{-2}$ with corresponding CE of 68.8%, 44.2% and 7.2%, respectively.

Figure 5c–e show the rate performance of $Li-CO_2$ batteries with the three cobalt sulfide cathodes. At the current density of 20 μA cm$^{-2}$, the mid-capacity polarizations of $CoS_2$, $CoS_{1.097}$, and $Co_9S_8$ cathodes are 0.74, 0.91, and 0.75 V, respectively. As the current density increased to 100 μA cm$^{-2}$, the overpotentials of the $CoS_2$ cathode are considerably lower than those of $CoS_{1.097}$ and $Co_9S_8$ cathodes, at only 1.19 V, while the overpotentials of the latter cathodes are ramped up significantly to 1.67 and 2.00 V, respectively. When the current density is reverted to 20 μA cm$^{-2}$, the overpotential of the $CoS_2$ cathode reduces to 0.33 V even lower than that of the first three cycles, in contrast to $CoS_{1.097}$ (1.67 V) and $Co_9S_8$ (1.63 V) cathodes, which exhibit inferior rate abilities (Supplementary Fig. 34). The cause of this phenomenon has been elucidated above and the same in long-term cycling. Under a current density of 20 μA cm$^{-2}$, the overpotential of $CoS_2$ is significantly reduced and retains stability in cycling as shown in Supplementary Fig. 35a, decreasing to 0.43 V after 400 h, which is also better than other sulfides catalysts (Supplementary Fig. 35b and Supplementary Table 1). Moreover, $CoS_2$ also maintains a long cycling life of over 700 h and maintains an overpotential lower than 1 V until battery failure (Supplementary Fig. 36 and Fig. 5f). In contrast, the overpotential of batteries with $CoS_{1.097}$ and $Co_9S_8$ cathodes exceeds 2 V only after 20 cycles. These

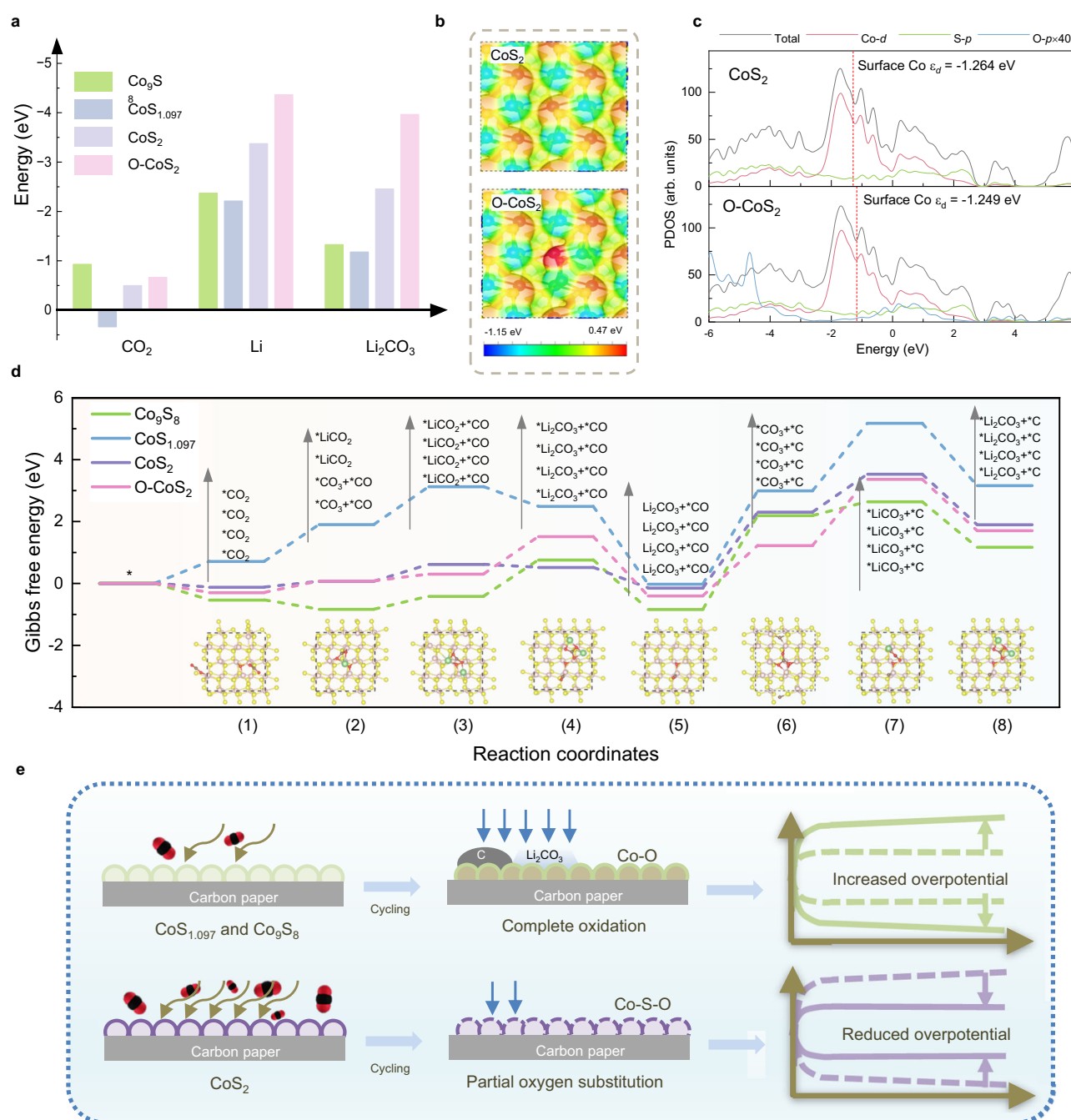

**Fig. 4 | DFT calculations and illustration of the reconstruction. a** The $CO_2$, Li, and $Li_2CO_3$ adsorption energies on $Co_9S_8$, $CoS_{1.097}$, $CoS_2$ and O-$CoS_2$. **b** Surface electrostatic potential diagrams of $CoS_2$ (up) and O-$CoS_2$ (down). **c** The projected density of states (PDOS) of $CoS_2$ (up) and O-$CoS_2$ (down); the inset red dotted line is the d-band center. **d** Gibbs free energy diagram of reaction pathways in Li-$CO_2$

batteries at U = 2.85 V on the four catalysts. The inset shows the top views of adsorption systems on O-$CoS_2$. **e** Illustration of the relationship between structural evolution and activity of $CoS_x$ in Li-$CO_2$ batteries. Source data are provided as a Source Data file.

contrasts for three cathodes are more visible in the selected cycles in Fig. 5g. Figure 5h shows Li-$CO_2$ cells based on $CoS_2$ electrodes with a solar-powered battery and a light-emitting diode (LED) array at day and night, which demonstrates its potential in Mars exploration and operation, where the atmosphere is 96% $CO_2$.

In conclusion, we have identified reconstructed motifs and unraveled the structure-activity relations of three cobalt (II) sulfides ($CoS_x$, x = 8/9, 1.097, and 2) in Li-$CO_2$ batteries by combining spectroscopy and DFT calculations. We uncover that most cobalt atoms in $Co_9S_8$ and $CoS_{1.097}$ coordinate with oxygen atoms after reconstruction, leading to

their deactivation and degradation in battery performance. In contrast, partial oxygen substitution with the Co-$S_4$-$O_2$ motif in $CoS_2$ contributes to the charge redistribution on cobalt atoms, thereby improving the catalytic ability. Reconstructed $CoS_2$ has a high energy efficiency (>80%) and superior stability during cycling with an overpotential of 0.43 V after 400 h in Li-$CO_2$ batteries. Our finding about active motifs and electronic structure features aids in understanding the high activity of sulfides and other transition compounds catalysts in Li-$CO_2$ batteries. We also expect our study can pave the way for the development of highly active and stable catalysts for metal-gas batteries.

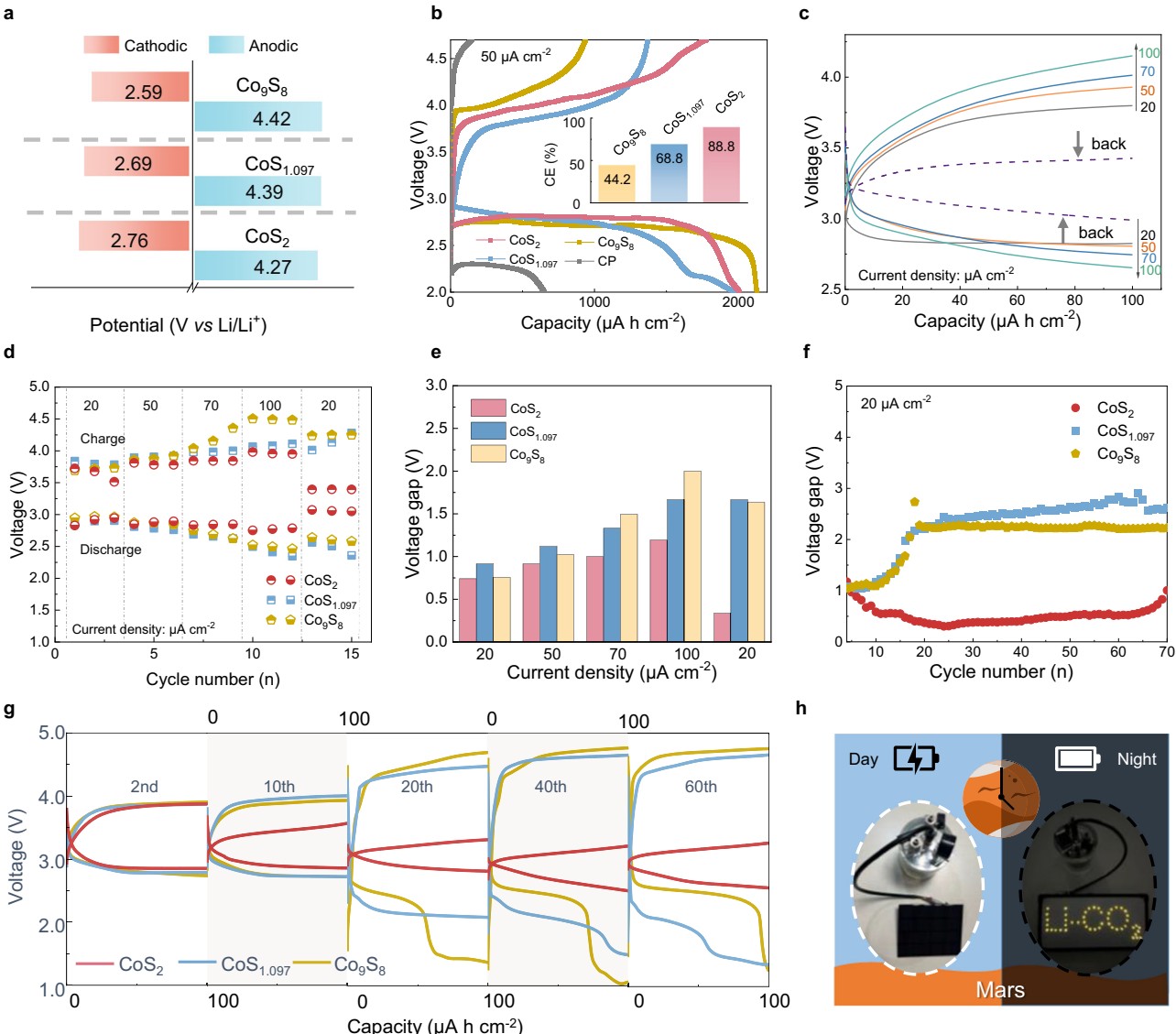

**Fig. 5 | Electrochemical performance. a** The onset potentials during discharge and charge and **b** fully discharging curves at a rate of $50\,\mu A\,cm^{-2}$ between 2 and 4.7 V (inset is the corresponding CE) for Li-$CO_2$ batteries with the three cathodes. **c** GDC profiles with a limited capacity of $100\,\mu A\,h\,cm^{-2}$ at different current densities for Li-$CO_2$ battery with the $CoS_2$ cathode. **d** Discharge and charge voltage and **e** overpotential at different current densities for the three cells. (The value of

overpotential is the average of three cycles with the same current density). **f** The voltage gap of the three cells for long-term cycling. **g** GDC profiles of selected cycles for the three cells. **h** Photo image of a solar-powered battery energy storage system based on Li-$CO_2$ batteries with $CoS_2$ cathodes at day and night. Source data are provided as a Source Data file.

## Method

### Synthesis of free-standing catalysts

Preparation of Co(OH)$_2$ nanosheets on carbon paper (Co(OH)$_2$/CP): CP (Toray, H-060) underwent an initial treatment at 700 °C for 10 min to enhance hydrophilicity. Electrodeposition was carried out using a three-electrode system, wherein the pre-treated CP (1 cm$^2$) served as the working electrode, a platinum mesh (1 cm$^2$) acted as the counter electrode, and a saturated calomel electrode (SCE) was used as the reference electrode. The electrolyte solution is 5 mM Co(NO)$_3$·6H$_2$O. Electrodeposition experiments were conducted using a CHI700E electrochemical workstation at a constant potential of −1.09 V (vs. SCE) for 15 s, followed by a rest period at 0 V (vs. SCE) for 15 s, lasting a total of 40 min. After electrodeposition, the deposited CP was rinsed several times with deionized water and subsequently dried in a vacuum at 60 °C for 12 h.

Preparation of $CoS_2$/CP: The Co(OH)$_2$/CP sample was placed at the center of the furnace tube in the quartz boat. 240 mg of S powder was

positioned upstream from the sample. The Co(OH)$_2$ nanosheets on the CP underwent a reaction with the S powder at 250 °C for 2 h. The temperature was ramped up at a rate of 10 °C per minute. Throughout this process, the environment within the tube was maintained as an Ar/H$_2$ mixture (5% H$_2$) flowing at a rate of 100 sccm.

Preparation of $CoS_{1.097}$/CP: The synthesis method for $CoS_{1.097}$/CP is similar to that for $CoS_2$/CP, except for the sulfur powder amount being 120 mg, the temperature being 300 °C, and the time being 2.5 h.

Preparation of $Co_9S_8$/CP: The synthesized $CoS_2$/CP was annealed at 300 °C for 1 h with a heating rate of 5 °C min$^{-1}$ under Ar/H$_2$(5%) (100 sccm) environment.

### Characterization

The morphologies were examined by SEM (Hitachi SU8010) and TEM (FEI Tecnai G2 F30). The cathodes after discharge and charge for SEM were extracted from coin cells, washed by TEGDME in the glovebox, and dried overnight in a vacuum at 60 °C before the test. XRD (Bruker

D8 Advance diffractometer) measurements were conducted to study the composition and structure. XPS spectra were collected using a Kratos AXIS Ultra DLD system to study the chemical states. The XPS results were evaluated with CasaXPS software and calibrated by shifting the main peak in the C $1s$ spectrum to 284.8 eV assigned to $sp^2$ carbon. A Pfeiffer QMG 250 DEMS (Germany) was used to measure the ratio of $CO_2$ evolution and $Li_2CO_3$ consumption during charge. The developed Li-$CO_2$ battery is in a homemade Swagelok battery cell (http://linglush.com). All electrodes for in situ DEMS test and titration are 1 cm*1 cm for ease of calculation. Lithium metal (diameter of 18 mm), borosilicate glass microfiber (diameter of 22 mm), and 100 μL 1 M lithium bis(trifluoromethane sulfone)imide (LiTFSI) in TEGDME were used as an anode, separator, and electrolyte, respectively. The battery for in situ analysis during charge is under a current density of 20 μA cm$^{-2}$ for 5 h after discharging with the same procedure, and an Ar flux is of 0.8 mL min$^{-1}$. For titration: cathodes were extracted from their respective Swagelok cells after discharge and charge and dried under vacuum without rinsing. They were then placed in a custom-built vessel (http://linglush.com). The 2 capillaries were attached to the DEMS apparatus and Ar through the vessel with a flux of 0.25 mL min$^{-1}$. After establishing a stable $CO_2$ and $O_2$ baseline, 1 mL of 3 M $H_3PO_4$ was injected into the vessel through a septa seal. The total amount of $CO_2$ evolved was calculated by integrating $CO_2$ flux. The $CO_2$ flux is determined as ppm ($CO_2$/Ar)*0.25 mL min$^{-1}$/22.4 L mol$^{-1}$. The DEMS cell was controlled by a LAND system.

## Electrochemical measurements

Coin cells (CR 2032) with several holes in the cathode were used to investigate the electrochemical performance of the Li-$CO_2$ batteries. Freestanding $CoS_2$/CP, $CoS_{1.097}$/CP, and $Co_9S_8$/CP (1 cm*1 cm) were directly used as the cathodes. Lithium metal, borosilicate glass microfiber (diameter of 18 mm), and 100 μL 1 M lithium bis(trifluoromethane sulfone)imide (LiTFSI) in TEGDME were used as an anode, separator, and electrolyte, respectively. The cells were assembled in an Ar-filled glovebox. Then them are transferred into chambers with pure $CO_2$ for electrochemical test. An electrochemical workstation Biologic SP150 and a LAND CT 2001A testing system were used to obtain the CV and discharge-charge curves, respectively. Because the potential of Li foil is easily affected by SEI and electrolytes, commercial $LiFePO_4$ electrodes are much more stable (aluminum foil single-side coated $LiFePO_4$ electrode, active material loading: 120 g/m$^2$) were used for the CV test[59]. ECSA measurement: The comparison of ECSA for cathodes was calculated based on $C_{dl}$, which is the double-layer capacitance. $C_{dl}$ was defined as $C_{dl} = (i_a-i_c)/2\upsilon$, $i_a$ is the anodic current, and $i_c$ is the cathodic current. $\upsilon$ is the scan rate of CVs in the non-faradaic region, an area between −0.26−−0.16 V of the open circuit potential (OCP). $C_{dl}$ was obtained by plotting $(i_a–i_c)/2$ as a function of $\upsilon$. All electrochemical tests are carried out in the room temperature.

## XAS measurements

The XAS spectra at the Co K-edges were recorded at the BL11B beamline of the Shanghai Synchrotron Radiation Facility (SSRF). The storage ring was operated at 3.5 GeV with a beam current of 200 mA in a top-up mode. The incident photons were monochromatized by a Si (111) double-crystal monochromator, with an energy resolution $\Delta E/E$-$1.4 \times 10^{-4}$. The spot size at the sample was ~200 μm × 250 μm (H × V). The XAS spectra of the samples at Co K-edges were calibrated by the Co reference foils (edge energy 7709 eV) collected in transition mode. The XAS spectra of the samples were collected in fluorescence mode, with a Lytle ionization chamber filled with Ar.

## XAFS data analysis

The data of XAFS were processed with ATHENA software implemented in the IFEFFIT software packages[60]. The raw data of XAFS were background subtracted from the overall absorption and then normalized

regarding the edge-jump step. Next, the k$^3$-weighted $\chi(k)$ data of Co K-edge were Fourier transformed to R space using a Hanning window (dk = 1.0 Å$^{-1}$) in k-space, which separates the contributions of different coordination shells to the EXAFS data. EXAFS of Co K-edge were Fourier transformed between 2.398 and 11.150 Å$^{-1}$. The quantitative structure parameters of Fe were obtained by least-squares fitting of EXAFS data with ATERMIS software in the IFEFFIT software packages. The fitting was according to the EXAFS Eq. (1):

$$\chi(k) = \sum_i \frac{N_i S_0^2 F_i(k)}{kR_i^2} \sin\left(2kR_i + \varphi_i(k)\right) e^{\frac{-2R_i}{\lambda(k)} - 2\sigma_i^2 k^2} \quad (7)$$

where $F_i(k)$, the effective scattering amplitude, $\lambda(k)$ the mean free path, and $\varphi_i(k)$, the effective scattering phase shift were theoretically calculated by the ab-intio code FEFF 6[49]. The fitting was conducted in R space with the single scattering path of the first coordination shells.

The wavelet transformations of k$^2$-weighted EXAFS of Co K-edge were performed in the k range between 2.50 and 11.50 Å$^{-1}$ with a k step of 0.05 Å$^{-1}$ and the R range between 0 and 6 Å with the hamaFortran program by using the Morlet wavelets[61]:

$$\varphi(k) = \frac{1}{\sqrt{2\pi}\sigma} e^{-\frac{k^2}{2\sigma^2}} (e^{i\omega k} - e^{-\frac{k^2}{2}}) \quad (8)$$

Where $\omega$ is the frequency and $\sigma$ is the half-width. To get a high resolution at the k-axis in the region of the first coordination shell, we chose $\omega = 3.5$ and $\sigma = 0.5$ for $CoS_2$ and $CoS_2-10$, and $\omega = 1$ and $\sigma = 1$ for CoO, respectively.

## Computation method

The first principles calculations were performed using the Vienna ab initio simulation package 6.4.0[62]. A plane-wave cutoff energy of 400 eV was used. The generalized gradient approximation proposed by Perdew, Burke, and Ernzerhof was used in the projector augmented wave method[63,64]. The computationally cost-effective Grimme's D3 scheme method for van der Waals (vdW) interactions was used to obtain a clear picture of weak interaction[65]. $Co_9S_8$(311), $CoS_{1.097}$(204), $CoS_2$(100) and O-$CoS_2$(100) were constructed. The detailed structural models can be found in Supplementary Data 1. Due to periodic boundary conditions, a vacuum separation of 20 Å between two neighboring monolayers was used. For the optimization and self-consistent calculations of surfaces, the Brillouin zone was sampled using the Monkhorst-Pack scheme with 0.05 of K-spacing value, respectively[66]. Ionic and electronic relaxations were performed by applying a convergence criterion of 0.05 eV/Å per ion and 10$^{-5}$ eV per electronic step, respectively. Here the dipole correction for slabs were considered in calculation. The Gibbs free energies (G) of each reaction intermediate were given by following Eq. (3):

$$G = E_{DFT} + E_{ZPE} - TS \quad (8)$$

where $E_{DFT}$, $E_{ZPE}$, T, and S are total energy by DFT calculations, the zero-point energy, temperature (298.15 K), and entropy, respectively. The entropies of other adsorbed molecules (T$\Delta$S) are calculated from the vibrational frequencies associated with the normal modes in the harmonic approximation.

The adsorption energy equals the energy of the adsorbed system minus the total energy of the substrate and the independent molecule or atom. The more negative the adsorption energy, the stronger the adsorption. By the Nernst equation, the calculated theoretical equilibrium potential $U_0$ is 2.85 V for 2 Li (s) + 3/2 $CO_2$ (g) → $Li_2CO_3$ (s) + 1/2 C (s), which is comparable to previous result[14,67].

## Data availability
All data that support the findings of this study are presented in the Manuscript and Supplementary Information, or are available from the corresponding author upon request. Source data are provided with this paper.

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

## Acknowledgements

This work was supported by the Joint Funds of the National Natural Science Foundation of China (U21A20174)[G.Z], Guangdong Innovative and Entrepreneurial Research Team Program (2021ZT09L197) [G.Z], Shenzhen Science and Technology Program (KQTD20210811090112002) [G.Z], Shenzhen Stabilization Support Program (WDZC20200824091903001) [G.Z], Guangdong Basic and Applied Basic Research Foundation (2020B0301030002) [H.-M.C], the start-up funds, Overseas Research Cooperation Fund, and Interdisciplinary Research and Innovation Fund of Tsinghua Shenzhen International Graduate School, and China Postdoctoral Science Foundation (No. 2020TQ0159) [B.C]. We also thank the staff in the BL11B beamline at Shanghai Synchrotron Radiation Facility (SSRF) for their technical assistance and Prof. Ji-Jing Xu and Mr. Dehui Guan from Jilin University for their support in the DEMS tests.

## Author contributions

Y.L., G.Z., and H.-M.C. conceived the idea for this project. Y.L. prepared the materials and electrochemical measurements. Z.Z. and B.L.L. carried out the XAS measurements and data analysis. D.W. conducted the DFT calculations and discussed the results. J.T. and B.L.L. performed the TEM test. B.C., B.Y.L., and R.M. assisted with battery testing and material synthesis. Y.L., D.W., Z.Z, G.Z., and H.-M.C. co-wrote the paper.

## Competing interests

The authors declare no competing interests.
