## [Peer Review File · Nature Communications]

Deciphering the Contributing Motifs of Reconstructed Cobalt (II) Sulfides Catalysts in Li-CO₂ BatteriesReviewer #1 (Remarks to the Author):

The goal of this work is to uncover the structure-activity relations of three cobalt (II) sulfides (CoS_x , $x=8/9$, 1.097, and 2) in Li-CO₂ batteries by combining spectroscopy and DFT calculations, and discovered that the varying oxidized states following reconstruction play a crucial role in determining battery performance. The conclusions of this article contribute to the understanding of the high activity of sulfides and other transition compounds catalysts in Li-CO₂ batteries. (Suggest Major Revision)

Here are a few suggested modifications:

(1) In this work, the CoS_x ($x=8/9$, 1.097, and 2) samples were synthesized by sulfidation of $\text{Co}(\text{OH})_2$ nanosheet arrays electrodeposited on pieces of carbon papers, so how to precisely control the elemental content and distribution uniformity?

(2) In Figure S2, what is the precise methodology for determining the numerical value of X based on the XRD results?

(3) Please proofread and make any necessary revisions to the text, such as, for Figure 1, Error bars of the voltage gap represents the standard deviation from three independent measurements, and for Reference 2.

(4) The variances in lattice spacing between the three samples in Fig.S4 are insignificant. Utilizing HRTEM diffraction patterns is advised for precise determination of the interplanar spacing.

(5) One of the key findings of the study is that CoS_2 undergoes partial oxidation, resulting in the formation of $\text{Co-S}_4\text{-O}_2$, unlike $\text{CoS}_{1.097}$ and Co_9S_8 , which undergo complete oxidation to CoO . However, the text only includes characterization of CoS_2 after 10 cycles, and there is no provision of X-ray spectra to establish its stability after extended cycling.

Reviewer #2 (Remarks to the Author):

In the submitted work, the authors consider Li-CO₂ batteries constructed with three different cobalt sulfide catalysts (Co_9S_8 , $\text{CoS}_{1.097}$, and CoS_2). While the authors have thoughtfully considered the structural evolution of the catalyst upon cycling of the batteries (which is a nice addition to the literature and is often overlooked), the evaluation of the overall battery reaction has not been given adequate attention or experimental efforts. As a result of this inattention, there are a number of significant issues that render the submitted work unsuitable for publication without significant additional experiments.

For instance, the authors assign the discharge process to $4\text{Li}+3\text{CO}_2\leftrightarrow 2\text{Li}_2\text{CO}_3+\text{C}$. Unfortunately, most of the discharge curves occur significantly above the thermodynamic potential for this process (2.80V), indicating that this discharge reaction is thermodynamically impossible. Moreover, there is little to no spectroscopic data to support this reaction pathway. On two of the three catalysts, there is no evidence provided for Li_2CO_3 or C formation. Moreover, the DEMS results which show CO₂ consumption upon discharge are under very different conditions/potentials than the rest of the experiments, rendering their support for the reaction pathway tenuous. Finally, the DEMS results provided upon charge (which are buried in the supporting information) demonstrate considerable irreversibility (>50%) on all three catalysts.

In order for the submitted work to be considered for publication, significant efforts are needed to fully characterize the discharge and charge reaction pathways. The review suggests using titration methods¹ to determine the quantity of formed Li_2CO_3 upon discharge and how much remains following charge, in addition to stronger spectroscopic evidence through techniques such as XRD or Raman (where any new peaks ascribed to products are clearly above the background noise of the measurement). Moreover, DEMS analysis must be performed under identical conditions/potentials to the other experiments to properly quantify gas consumption/evolution during cycling. Finally, clear spectroscopic evidence that the redox state of the Cobalt does not change between the discharged and charged states is essential to rule out the role of transition metal redox in the observed electrochemistry. Additional detailed comments are provided below.

Major comments:

- Many of the discharge plateaus occur at voltages greater than the thermodynamic potential of the proposed reaction ($4\text{Li}+3\text{CO}_2\leftrightarrow 2\text{Li}_2\text{CO}_3+\text{C}$ at 2.80 V). For instance, all discharge plateaus in

Figure 1c-e are >2.9V. It is therefore not thermodynamically possible that the formation of Li_2CO_3 and C during discharge is the correct overall reaction, as even with a catalyst, the potential must be below the thermodynamic potential. As a result, the reviewer has very serious concerns about the presented analyses, which assume that the formation of Li_2CO_3 and C occur during discharge. This is also observed in the CV shown in Figure S23, where the onset potential on CoS_2 is >3V. This is especially concerning given the small discharge capacities used in most experiments, as well as the presence of cobalt, which can be redox active.

- The provided spectroscopic evidence that the dominant reaction is $4\text{Li} + 3\text{CO}_2 \leftrightarrow 2\text{Li}_2\text{CO}_3 + \text{C}$ is very weak. For instance, the XRD diffraction patterns in Figure 2b show essentially no unambiguous peaks associated with Li_2CO_3 for either CoS_2 nor Co_9S_8 . No other spectroscopic evidence is provided to evaluate the formation of Li_2CO_3 during discharge on these catalysts.

- The charging results from DEMS in Figure S7 show very poor reversibility (<~50% of expected CO_2 evolution) for all of the prepared catalysts. The reviewer is very concerned that this result was buried in the SI and not acknowledged adequately in the text.

- The voltage profiles of DEMS cells differ significantly from those of non-DEMS cells in Figure 1c-e and 2a. For instance, the discharge profile of both $\text{CoS}_1.097$ and Co_9S_8 drops to 2V for the DEMS cells, but maintains a plateau ~2.9V in Figure 1d-e. Can the authors please rationalize this difference? Moreover, there is inadequate information in the methods section to figure out the current density applied to the DEMS cells (current is given as 400 μA , but no area is given).

- The DFT reaction pathway shows a rate determining step with an energy barrier of >2 eV. If this pathway were correct, there should be massive overpotential during the discharge process, which is not observed – highlighting a significant disagreement between the DFT computed pathway and the observed electrochemistry.

Minor comments:

- Introduction, line 3 – please use “humankind” or similar, instead of the gendered “mankind”
- Caption of Figure 2 contains errors (lists capacity as 200 mAh/cm² instead of 200 $\mu\text{Ah}/\text{cm}^2$)
- X axis of Figure 2a is confusing as the capacity increases with both discharge and charge, instead of reversing directions upon charging as is the convention.
- Please ensure the areal charge/discharge rate is clearly labelled in all figures/captions
- Page 10, line 3- “partial oxygen substitution effectively improves the adsorption strengths” should be “increases the adsorption strengths”. You have no evidence as of yet to show that increased adsorption strength is more favorable catalytically.

References cited in this review:

1. McCloskey, B. D. et al. Combining Accurate O_2 and Li_2O_2 Assays to Separate Discharge and Charge Stability Limitations in Nonaqueous Li- O_2 Batteries. *J. Phys. Chem. Lett.* 4, 2989–2993 (2013).

Reviewer #3 (Remarks to the Author):

All of the contents are well-organized in the manuscript, and there are many meaningful results and insights. Therefore, I recommend accepting this manuscript for publication after simple revision on the following comments;

1. It would be better to add the electrochemical performances of just carbon-cloth-based Li- CO_2 battery without the catalyst, to show the catalytic effect of cobalt sulfate more clearly.
2. The surface area and pore distribution can affect the electrochemical performances of Li- CO_2 battery. Thus, comparison of the surface area and pore distribution among the CoS_2 , $\text{CoS}_1.097$, and Co_9S_8 -based electrode are required.
3. The authors claimed that oxidation on CoS_2 is terminated with Co-S₄- O_2 motifs. Since Li- CO_2 battery system cannot provide O_2 molecule directly, it would be required to explain formation process of Co-S₄- O_2 motif and O_2 source in more detail.
4. It would be better if this concept is not limited to cobalt-based sulfides and a detailed discussion and perspective are added on the applicability to various metal sulfides.
5. In Computational details, it seems $\text{Co}_9\text{S}_8(311)$, $\text{CoS}_1.097(204)$, $\text{CoS}_2(100)$ and O- $\text{CoS}_2(100)$ means the plane information of the materials. Is there any reason to use the different type of the plane rather than same type of the plane for DFT calculation?

Response to Reviewer #1

Comments:

The goal of this work is to uncover the structure-activity relations of three cobalt (II) sulfides (CoS_x , $x=8/9$, 1.097, and 2) in Li- CO_2 batteries by combining spectroscopy and DFT calculations, and discovered that the varying oxidized states following reconstruction play a crucial role in determining battery performance. The conclusions of this article contribute to the understanding of the high activity of sulfides and other transition compounds catalysts in Li- CO_2 batteries. (Suggest Major Revision)

Response. Thank you very much for the positive recommendations.

Here are a few suggested modifications:

(1) In this work, the CoS_x ($x=8/9$, 1.097, and 2) samples were synthesized by sulfidation of $\text{Co}(\text{OH})_2$ nanosheet arrays electrodeposited on pieces of carbon papers, so how to precisely control the elemental content and distribution uniformity?

Response 1. Thanks for the question. In our work, the sulfidation is in the reductive atmosphere which contains 5% H_2 in Ar. Usually, the valence states of sulfur powder can be reduced to lower by increasing reaction temperature and time. According to this, we have successfully synthesized CoS_2 (250 °C for 2 h), $\text{CoS}_{1.097}$ (300 °C for 2.5 h), and Co_9S_8 (anneal CoS_2 in Ar/ H_2) by sulfidation of $\text{Co}(\text{OH})_2$ nanosheet arrays electrodeposited on pieces of carbon papers ($\text{Co}(\text{OH})_2/\text{CP}$). To realize the distribution uniformity, the $\text{Co}(\text{OH})_2/\text{CP}$ is placed vertically in the flow direction, and the size is strictly controlled to 1 cm*1 cm. The characterizations show no presence of heterophase after sulfidation, indicating the validity of our method. This is also universally applicable to the synthesis of other sulfides. As shown in Figure R1, nickel sulfides with different Ni:S ratios are synthesized in this way.

Figure R1 XRD patterns of (a) NiS₂, (b) NiS, and (c) Ni₃S₂.

(2) In Figure S2, what is the precise methodology for determining the numerical value of X based on the XRD results?

Response 2. Thanks for the question. In this paper, we used CoS_x (X=2, 1.097, and 8/9) to represent the three cobalt sulfides for convenience. For cobalt (II) sulfides, the numerical value of X is not continuous and cannot be valued subjectively. The common values are 2, 8/9, and around 1, the structures of which are quite different. CoS₂ is the pyrite structure, while for Co₉S₈, eight Co atoms are in tetrahedral holes and the ninth one is in an octahedral hole in a cubic close-packed sulfur array. Cobalt monosulfide has a NiAs structure and exists with excess sulfur atoms (Co_{1-x}S) in some specific forms. Such differences in structures of the three sulfides can be distinguished by XRD characterizations. In our work, the locations of strong peaks of synthesized samples are in accordance with PDF#41-1471, #19-0366, and #02-1459. Therefore, the values of X are determined as 2, 1.097, and 8/9.

(3) Please proofread and make any necessary revisions to the text, such as, for Figure 1, Error bars of the voltage gap represents the standard deviation from three independent measurements, and for Reference 2.

Response 3. Thank you for your kind suggestion. We have carefully gone through the manuscript, but haven't found the error in reference 2. Instead, we found errors in the caption of Figure 2. The errors have been revised in the revision.

(4) The variances in lattice spacing between the three samples in Fig.S4 are insignificant.

Utilizing HRTEM diffraction patterns is advised for precise determination of the interplanar spacing.

Response 4. Thank you for your kind suggestion. We used the Fast Fourier Transform (FFT) of the HRTEM image to visualize the diffraction patterns and determined the d-spacing values by measuring the distance between two symmetric diffraction points in the FFT patterns as shown in Figure R2. The values are 0.273 nm, 0.292 nm, and 0.297 nm for CoS_2 , $\text{CoS}_{1.097}$ and Co_9S_8 , respectively. We have updated Figure S5 in the revised SI.

Figure R2. TEM images of (a) CoS_2 , (b) $\text{CoS}_{1.097}$, and (c) Co_9S_8 . The insets are the corresponding Fast Fourier Transform (FFT) patterns.

(5) One of the key findings of the study is that CoS_2 undergoes partial oxidation, resulting in the formation of $\text{Co-S}_4\text{-O}_2$, unlike $\text{CoS}_{1.097}$ and Co_9S_8 , which undergo complete oxidation to CoO . However, the text only includes characterization of CoS_2 after 10 cycles, and there is no provision of X-ray spectra to establish its stability after extended cycling.

Response 5. Thank you for your comment. From Figure S8, the $\text{S } 2p_{1/2}$ and $\text{S } 2p_{3/2}$ peaks of CoS_2 after 5 cycles are located in the same position as the peaks after 10 cycles. Therefore, we inferred that the chemical states of CoS_2 maintain stability during battery cycling. Regarding the feedback and support for this point, we performed XPS characterization for CoS_2 after extended cycling as shown in Figure R3. Figure R3a shows the existence of Co-S binding after 20 and 30 cycles on CoS_2 . The location of $\text{S } 2p$ peaks on the CoS_2 after 20 and 30 cycles are also the same as that after 5 and 10 cycles (Figure R3b). These results suggest similar chemical states of reconstructed CoS_2 during cycling, which confirms the stability of $\text{Co-S}_4\text{-O}_2$. We updated Figures S8

and 9 and modified the sentences on Page 13 in the revised SI as follows:

CoS₂, on the other hand, exhibits no significant decline in the intensity of S 2p_{1/2} and S 2p_{3/2} peaks. Instead, the peaks shift ~0.6 eV to lower binding energy after 5 cycles, without moving after 10, 20, and 30 cycles. These results indicate that S with less than the full coordination is in CoS₂, and the structure of reconstructed CoS₂ remains stable during battery operation.

Figure R3. (a) Co 2p and (b) S 2p XPS spectra of CoS₂ after cycling.

Response to Reviewer #2

Comments:

In the submitted work, the authors consider Li-CO₂ batteries constructed with three different cobalt sulfide catalysts (Co₉S₈, CoS_{1.097}, and CoS₂). While the authors have thoughtfully considered the structural evolution of the catalyst upon cycling of the batteries (which is a nice addition to the literature and is often overlooked), the evaluation of the overall battery reaction has not been given adequate attention or experimental efforts. As a result of this inattention, there are a number of significant issues that render the submitted work unsuitable for publication without significant additional experiments.

For instance, the authors assign the discharge process to $4\text{Li} + 3\text{CO}_2 \leftrightarrow 2\text{Li}_2\text{CO}_3 + \text{C}$. Unfortunately, most of the discharge curves occur significantly above the thermodynamic potential for this process (2.80 V), indicating that this discharge reaction is thermodynamically impossible. Moreover, there is little to no spectroscopic data to support this reaction pathway. On two of the three catalysts, there is no evidence provided for Li₂CO₃ or C formation. Moreover, the DEMS results which show CO₂ consumption upon discharge are under very different conditions/potentials than the rest of the experiments, rendering their support for the reaction pathway tenuous. Finally, the DEMS results provided upon charge (which are buried in the supporting information) demonstrate considerable irreversibility (>50%) on all three catalysts.

In order for the submitted work to be considered for publication, significant efforts are needed to fully characterize the discharge and charge reaction pathways. The review suggests using titration methods¹ to determine the quantity of formed Li₂CO₃ upon discharge and how much remains following charge, in addition to stronger spectroscopic evidence through techniques such as XRD or Raman (where any new peaks ascribed to products are clearly above the background noise of the measurement).

Moreover, DEMS analysis must be performed under identical conditions/potentials to the other experiments to properly quantify gas consumption/evolution during cycling. Finally, clear spectroscopic evidence that the redox state of the Cobalt does not change between the discharged and charged states is essential to rule out the role of transition metal redox in the observed electrochemistry. Additional detailed comments are provided below.

Response. Thank you very much for the thorough review of our manuscript. We have carefully considered the feedback regarding the reaction pathway in Li-CO₂ batteries. As per the suggestions, we have conducted additional experiments and made revisions, which are outlined below.

Major comments

- Many of the discharge plateaus occur at voltages greater than the thermodynamic potential of the proposed reaction ($4\text{Li}+3\text{CO}_2\leftrightarrow 2\text{Li}_2\text{CO}_3+\text{C}$ at 2.80 V). For instance, all discharge plateaus in Figure 1c-e are >2.9V. It is therefore not thermodynamically possible that the formation of Li₂CO₃ and C during discharge is the correct overall reaction, as even with a catalyst, the potential must be below the thermodynamic potential. As a result, the reviewer has very serious concerns about the presented analyses, which assume that the formation of Li₂CO₃ and C occur during discharge. This is also observed in the CV shown in Figure S23, where the onset potential on CoS₂ is >3V. This is especially concerning given the small discharge capacities used in most experiments, as well as the presence of cobalt, which can be redox active.

Response. Thank you very much for the comments. We understand the reviewer's concerns about the reaction pathways. Regarding this, we have retested the CV performance in CO₂ and Ar atmosphere with LiFePO₄ instead of Li foil as the reference electrode. Li foil is a quasi-reference electrode, whose potential is affected by SEI, electrolytes, and other environmental factors, while LiFePO₄ is much more stable during battery operation and unsusceptible to different atmospheres.¹ As shown in Figure R4, all batteries exhibit featureless curves in the Ar atmosphere while obvious oxidation and reduction peaks in the CO₂ atmosphere, which indicates the

electrochemical inactivity of sulfides and carbon papers without CO₂ at the range of 2.2~4.7 V. Besides, Co 2p XPS spectra (Figure R5) of three cathodes show that the location of Co-O and Co-S binding are unmovable during discharge and charge. Even though the intensity of Co-S binding decline in charged Co₉S₈ and CoS_{1.097} after cycling, the XAS results in Figures 2a and b that the energies of absorption edges are still lower than that of CoO, suggesting the valence states of Co maintain +2. These results indicate that the valence states of cobalt haven't changed after discharge and charge, and battery reactions mainly rely on CO₂ for the cathodes.

Based on this, we consider the reactions for Li and CO₂ as the following reactions (1)-(4), two of which ((2) and (4)) are thermodynamically possible. Although no obvious signal of Li₂C₂O₄ has been observed, the reaction (2) cannot be exclusive arbitrarily because Li₂C₂O₄ may disproportionate to Li₂CO₃ under discharge potential. Also, oxygen species formation is possible under battery operation. However, verifying these two assumptions with cogent results is quite difficult with current characterization technologies. The charge-to-mass of Li₂CO₃ is 2e⁻/Li₂CO₃ in all reactions (1)-(4), so the quantities of formed Li₂CO₃ by titration can't be used to determine the discharge reaction. The e⁻/CO₂ ratio is 2 for the two reactions which differ from reactions (1) and (3). Nonetheless, the consumption of CO₂ is very hard to detect in identical conditions by *in situ* DEMS test, due to the signal easily drowning in the background noise, especially under such a small current density. The characterization of oxygen species is more challenging for the disturbance from environmental oxygen and perishability for high activity.

In this work, we mainly focus on the relationship between electrochemical performance and catalyst structure during battery cycling. The differences are mainly in the change of charge voltage of the three sulfides after reconstruction as shown in Figure 1c-e. We also have found this phenomenon that higher discharge voltage (>2.8 V) with discharge product Li₂CO₃ in the previous reports on different catalysts in Table S1.^{2, 3, 4 5, 6, 7, 8, 9} This suggests it is a phenomena in this field, calling for profound thoughts and massive efforts. We apologize for the non-comprehensive analysis and statements for discharge reactions, and revised them on Page 8 in the revised manuscript

and added Figure S12 in the revised SI.

Figure R4. CV curves of (a) Co₉S₈, (b) CoS_{1.097}, (c) CoS₂, and (d) carbon paper in CO₂ and Ar atmosphere.

Figure R5. XPS results of Co 2p of (a) Co₉S₈, (b) CoS_{1.097}, and (c) CoS₂ after discharge and charge.

- The provided spectroscopic evidence that the dominant reaction is $4\text{Li}+3\text{CO}_2\leftrightarrow 2\text{Li}_2\text{CO}_3+\text{C}$ is very weak. For instance, the XRD diffraction patterns in

Figure 2b show essentially no unambiguous peaks associated with Li_2CO_3 for either CoS_2 nor Co_9S_8 . No other spectroscopic evidence is provided to evaluate the formation of Li_2CO_3 during discharge on these catalysts.

Response. Thank you very much for the comments. We are sorry for our ambiguous peaks of Li_2CO_3 in XRD patterns, which are affected by the crystallinity and quantity of discharged products and background noise of carbon papers. To avoid the disturbance, we have performed Raman spectroscopy of CoS_2 and Co_9S_8 before and after discharge, as shown in Figure R6. The new peaks at 1080 cm^{-1} emerged on both Co_9S_8 and CoS_2 after discharge, which are assigned to Li_2CO_3 . These results have been added as Figure S13 in the revised SI.

Figure R6. Raman spectra of cathodes before and after discharge. The reference spectrum for Li_2CO_3 and carbon paper is also shown.

- The charging results from DEMS in Figure S7 show very poor reversibility (<~50% of expected CO_2 evolution) for all of the prepared catalysts. The reviewer is very concerned that this result was buried in the SI and not acknowledged adequately in the

text.

Response. We sincerely thank the reviewer for bringing our attention to the omission of charge reaction. By now, there are two possible charge reaction pathways:

We first study the charge reaction by calculating the ratio of Li_2CO_3 and CO_2 , which is ~ 0.67 for reaction (5) and 1 for reaction (6). The quantity of reacted Li_2CO_3 during the charge is determined by titration methods as the reviewer suggested. Titrating Li_2CO_3 solution with a certain concentration is performed as an external standard to diminish the measurement error in Figures R7a-c and Table R1. Figure R7d shows that the measurement value is approximately linear with the theoretical amount as a function of $y = 0.89847 * x - 0.05068$. y is the measurement value of CO_2 evolution and x is the theoretical amount of Li_2CO_3 . Based on this, the quantities of formed and residual Li_2CO_3 during discharge and charge under a current density of $20 \mu\text{A cm}^{-2}$ on the three cathodes are shown in Figure R8 and Table R2. *In situ* DEMS during charge with identical current density are performed in Figure R9, in which only CO_2 ($m/z=44$) generation can be observed on all cathodes. The amount of CO_2 generation on CoS_2 is much higher than that on Co_9S_8 and $\text{CoS}_{1.097}$, and the numerical results are presented in Table R3. Among them, the ratio of Li_2CO_3 to CO_2 of $\text{CoS}_{1.097}$ is 0.96, close to 1, indicating that reaction (6) may mostly happen during charge. Even though no signal of O_2 ($m/z=32$) has been observed, oxygen species generation is commonly possible, threatening the catalyst's durability. Consequently, $\text{CoS}_{1.097}$ is oxidized to CoO during cycling and affects its electrochemical performance. The ratio of Co_9S_8 and CoS_2 is ~ 0.76 and ~ 0.63 respectively, close to that of the reaction (5). However, the conversion efficiency of Li_2CO_3 on Co_9S_8 is only 18.4 % much lower than the other cathodes, indicating oxidation reactions mainly happened to supply capacity. The XAS results in Figures 2a-c show that the valence states of Co in three sulfides maintain +II after 10 cycles, excluding the possibility of Co contribution to the charge capacity. Instead, the sulfur oxidation may be responsible for the charge capacity of Co_9S_8 , as the decreased intensity of Co-S binding in Figure S8a. We also can't exclude the possibility of

electrolyte decomposition that supply the capacity and oxidize the catalysts. By contrast, the higher charge efficiency of CoS₂ benefits its reconstruction to Co-S₄-O₂ instead of complete oxidation.

We have added these results as Figures S14-16 and Tables S3-5 in the revised SI, updated Figure 3 and discussed it in the revised manuscript on Pages 9-10, as “*After charge.... instead of complete oxidation*”.

The details of the calculation and test were also added in Characterization on Page 3 in the revised SI:

*“A Pfeiffer QMG 250 DEMS (Germany) was used to measure the ratio of CO₂ evolution and Li₂CO₃ consumption during charge. The developed Li-CO₂ battery is in a homemade Swagelok battery cell (<http://linglush.com>). In situ analysis during charge was performed with an Ar flux of 0.8 mL min⁻¹ at a current density of 20 μA cm⁻² for 5h. For titration: cathodes after discharge and charge were extracted from their respective cells and dried under vacuum without rinsing. They were then placed in a custom-built vessel (<http://linglush.com>), The 2 capillaries were attached to the DEMS apparatus and Ar through the vessel with a flux of 0.25 mL min⁻¹. After establishing a stable CO₂ and O₂ baseline, 1 mL of 3M H₃PO₄ was injected into the vessel through a septa seal. The total amount of CO₂ evolved was calculated by integrating CO₂ flux. The CO₂ flux is determined as ppm (CO₂/Ar)*0.25 mL min⁻¹/22.4 L mol⁻¹. The DEMS cell was controlled by a LAND system.”*

Figure R7. The CO₂ ($m/z=44$) generation after titrating (a) 0.1 mL, (b) 0.05 mL, and (c) 0.025 mL Li₂CO₃ solution with a certain concentration of 5.25 mg mL⁻¹. (d) The relationship between measurement value of CO₂ and theoretical amount of Li₂CO₃.

Figure R8. The CO₂ (m/z=44) generation after titrating (a) discharged and (b) charged Co₉S₈; (c) discharged and (d) charged CoS_{1.097}; (e) discharged and (f) charged CoS₂.

Figure R9. The gas ($m/z=44$, 28, and 32) generation during the charge on (a) Co_9S_8 , (b) $\text{CoS}_{1.097}$, and (c) CoS_2 .

Table R1. The measurement value and theoretical amount of CO_2 evolution.

Amount of Li_2CO_3 (μg)	Theoretical amount of Li_2CO_3 (μmol)	Measurement value of CO_2 evolution (μmol)
525	7.09	6.29
262.5	3.55	3.20
131.25	1.77	1.48

Table R2. The quantities of Li_2CO_3 after discharge and charge on the three cathodes.

Cathode	Measurement value of CO_2 evolution (μmol)		Amount of Li_2CO_3 (μmol)		Conversion efficiency of Li_2CO_3
	After discharge	After charge	After discharge	After charge	
Co_9S_8	1.46	1.18	1.68	1.37	18.4%

CoS _{1.097}	0.91	0.55	1.06	0.61	42.4%
CoS ₂	1.24	0.61	1.44	0.66	54.2%

The conversion efficiency of Li₂CO₃ is defined as $(N_{ad}-N_{ac})/N_{ad}$.

N_{ad} is the amount of Li₂CO₃ after discharge, and N_{ac} is the amount of Li₂CO₃ after charge.

Table R3 The amount of CO₂ evolution and Li₂CO₃ consumption during charge.

Cathode	Amount of CO ₂ evolution (μmol)	Amount of Li ₂ CO ₃ consumption (μmol)	Li ₂ CO ₃ /CO ₂
Co ₉ S ₈	0.41	0.31	~0.76
CoS _{1.097}	0.47	0.45	~0.96
CoS ₂	1.24	0.78	~0.63

- The voltage profiles of DEMS cells differ significantly from those of non-DEMS cells in Figure 1c-e and 2a. For instance, the discharge profile of both CoS_{1.097} and Co₉S₈ drops to 2 V for the DEMS cells, but maintains a plateau ~2.9V in Figure 1d-e. Can the authors please rationalize this difference? Moreover, there is inadequate information in the methods section to figure out the current density applied to the DEMS cells (current is given as 400 uA, but no area is given).

Response. Thank you very much for the comments. The area of cathodes used for previous DEMS tests is 3.14 cm². The corresponding current density is ~127.4 μA cm⁻², more than five times the current density used in Figure 1c-e. Therefore, the discharge voltages of all cathodes for the DEMS cells are lower than those in non-DEMS cells. Figure R10 shows the cathodes in DEMS cells with a current density of 20 μA cm⁻², which shows little difference with non-DEMS cells in the same condition.

Figure R10. Discharge profiles of Co_9S_8 , $\text{CoS}_{1.097}$, and CoS_2 in DEMS cell.

- The DFT reaction pathway shows a rate determining step with an energy barrier of >2 eV. If this pathway were correct, there should be massive overpotential during the discharge process, which is not observed – highlighting a significant disagreement between the DFT computed pathway and the observed electrochemistry.

Response. Thank you very much for the comments. As the previous reports, the reaction steps between $^*\text{CO}$ and CO_2 are complex and still unclear. By now, a simplified reaction step has been proposed and used: $^*\text{CO}$ reacts with CO_2 to directly form $^*\text{CO}_3$ and $^*\text{C}$, and $^*\text{CO}_3$ then reacts with Li to form the next $^*\text{Li}_2\text{CO}_3$.^{10, 11} We also used this simplified reaction path to help study the possible reaction energy profiles, which cannot fully equal to actual reaction kinetics. In this case, developing time-resolved characterization to capture more reaction intermediates is difficult but meaningful to understand the reaction pathways and get a more accurate calculation result for future studies in Li- CO_2 batteries.

Minor comments:

- Introduction, line 3 – please use “humankind” or similar, instead of the gendered “mankind”

- Caption of Figure 2 contains errors (lists capacity as 200 mAh/cm² instead of 200 uAh/cm²)
- X axis of Figure 2a is confusing as the capacity increases with both discharge and charge, instead of reversing directions upon charging as is the convention.
- Please ensure the areal charge/discharge rate is clearly labelled in all figures/captions
- Page 10, line 3- “partial oxygen substitution effectively improves the adsorption strengths” should be “increases the adsorption strengths”. You have no evidence as of yet to show that increased adsorption strength is more favorable catalytically.

Response. Thank you for the comments. We have carefully gone through the manuscript and revised these errors in the revised version.

References cited in this review:

1. McCloskey, B. D. et al. Combining Accurate O₂ and Li₂O₂ Assays to Separate Discharge and Charge Stability Limitations in Nonaqueous Li–O₂ Batteries. *J. Phys. Chem. Lett.* 4, 2989–2993 (2013)

Response to Reviewer #3

Comments:

All of the contents are well-organized in the manuscript, and there are many meaningful results and insights. Therefore, I recommend accepting this manuscript for publication after simple revision on the following comments;

Response. Thank you very much for the positive recommendations

1. It would be better to add the electrochemical performances of just carbon-cloth-based Li-CO₂ battery without the catalyst, to show the catalytic effect of cobalt sulfate more clearly.

Response 1. Thank you for your kind suggestion. We have performed the CV and GDC for CP in the CO₂ atmosphere. As shown in Figure R11a, CP shows insignificant redox and oxidation peaks in both CO₂ and Ar atmosphere. Besides, CP shows a large overpotential (>2.7 V) and poor reversibility with a Coulombic efficiency of 7.2% in Figure R11b, indicating its poor catalytic ability in Li-CO₂ batteries. We have added some sentences on Pages 13-14 and updated Figures 5a and b in the revised manuscripts, and Figure S28 in the revised SI.

Figure R11 (a) The CV curves in the CO₂ and Ar atmosphere and (b) GDC profile at a current density of 50 μA cm⁻² of CP.

2. The surface area and pore distribution can affect the electrochemical performances of Li-CO₂ battery. Thus, comparison of the surface area and pore distribution among

the CoS₂, CoS_{1.097}, and Co₉S₈-based electrode are required.

Response 2. Thank you for your kind suggestion. The three sulfides are all synthesized with the same precursor, and the morphologies show no significant change after sulfidation in Figure S3. To evaluate the electrochemical surface area (ECSA) of the three samples, CV scans in the non-Faradaic region as shown in Figures R12a-c. The values of calculated electrochemical double-layer capacity (C_{dl}) in Figure R12d are very similar, which of CoS₂, CoS_{1.097}, and Co₉S₈ are 6.82, 9.75, and 8.44 mF cm⁻². These results indicate that the difference in the electrochemical performance of the sulfides is not due to their surface area. We have added the results in Figure S4 in the revised SI and mentioned this in the manuscript on Page 6:

Figure S4 also confirms that the electrochemical surface active area (ECSA) of the three cathodes are similar, ruling out their influence on the following electrochemical test.

The details of the calculation and test were also added in Electrochemical Measurement on Page 3-4 in the revised SI:

ECSA measurement: The comparison of ECSA for cathodes was calculated based on C_{dl} , which is the double-layer capacitance. C_{dl} was defined as $C_{dl}=(i_a-i_c)/2v$, i_a is the anodic current, and i_c is the cathodic current. v is the scan rate of CVs in the non-faradaic region, an area between -0.26~-0.16V of the open circuit potential (OCP). C_{dl} was obtained by plotting $(i_a-i_c)/2$ as a function of v .

Figure R12 ECSA test of (a) CoS₂, (b) CoS_{1.097}, and (c) Co₉S₈. (d) C_{dl} of the three cathodes.

3. The authors claimed that oxidation on CoS₂ is terminated with Co-S₄-O₂ motifs. Since Li-CO₂ battery system cannot provide O₂ molecule directly, it would be required to explain formation process of Co-S₄-O₂ motif and O₂ source in more detail.

Response 3. Thank you for your kind suggestion. Even though Li-CO₂ battery system cannot provide O₂ molecules directly, oxygen species generation under battery operation and cycling oxidize the catalyts.^{12, 13} During charge, part of Li₂CO₃ may decompose with the parasitic reaction $\text{Li}_2\text{CO}_3 \rightarrow 2\text{Li} + \text{CO}_2 + 1/2\text{O}_2 / \text{O}^\cdot$. We also can't exclude O in the electrolytes and its decomposition for the high charge potential and efficiency lower than 100% in Li-CO₂ batteries. We added some discussion in the revised manuscripts on the Pages 9-10 to explain the reconstruction process in more detail as:

“By now, there are two possible charge reaction pathways.... its reconstruction to Co-S₄-O₂ instead of complete oxidation.”

4. It would be better if this concept is not limited to cobalt-based sulfides and a detailed discussion and perspective are added on the applicability to various metal sulfides.

Response 4. Thank you for your kind suggestion. We have considered the applicability to other metal sulfides, such as nickel. As shown in Figures R13a and b, we synthesized nickel sulfides on carbon paper in the similar way. However, even though NiS₂ is much more stable than NiS, the overpotential does not reduce like CoS₂ during cycling (Figure R13c). Therefore, we assumed that cobalt may affect the reconstruction in CoS₂ under battery operation and the underlying law for structure evolution of various sulfides should be studied in the future.

Figure R13 XRD patterns of (a) NiS and (b) NiS₂. (c) Time-voltage curves at 20 μA cm⁻² of NiS and NiS₂.

5. In Computational details, it seems Co₉S₈(311), CoS_{1.097}(204), CoS₂(100) and O-CoS₂(100) means the plane information of the materials. Is there any reason to use the different type of the plane rather than same type of the plane for DFT calculation?

Response 5. Thank you for your question. The three sulfides have very different

structures. CoS_2 is the pyrite structure, while $\text{CoS}_{1.097}$ is a NiAs structure. Co_9S_8 has the structure that eight Co atoms are in tetrahedral holes and the ninth one is in an octahedral hole in a cubic close-packed sulfur array. It is unreasonable to choose the same plane for different crystal structures. Therefore, we chose the planes for the three sulfides that we commonly observed in TEM as shown in Figure S5.

Reference

1. Murdock BE, Armstrong CG, Smith DE, Tapia-Ruiz N, Toghiani KE. Misreported non-aqueous reference potentials: The battery research endemic. *Joule* **6**, 928-934 (2022).
2. Lin J, *et al.* Boosting Energy Efficiency and Stability of Li-CO₂ Batteries via Synergy between Ru Atom Clusters and Single-Atom Ru-N₄ sites in the Electrocatalyst Cathode. *Adv Mater* **34**, 2200559 (2022).
3. Fan L, *et al.* Biaxially Compressive Strain in Ni/Ru Core/Shell Nanoplates Boosts Li-CO₂ Batteries. *Adv Mater* **34**, 2204134 (2022).
4. Qiao Y, *et al.* Synergistic effect of bifunctional catalytic sites and defect engineering for high-performance Li-CO₂ batteries. *Energy Storage Materials* **27**, 133-139 (2020).
5. Qiao Y, *et al.* Transient, in situ synthesis of ultrafine ruthenium nanoparticles for a high-rate Li-CO₂ battery. *Energy Environ Sci* **12**, 1100-1107 (2019).
6. Jin Y, Chen F, Wang J, Johnston RL. Tuning electronic and composition effects in ruthenium-copper alloy nanoparticles anchored on carbon nanofibers for rechargeable Li-CO₂ batteries. *Chem Eng J* **375**, 121978 (2019).
7. Qiao Y, *et al.* 3D-Printed Graphene Oxide Framework with Thermal Shock Synthesized Nanoparticles for Li-CO₂ Batteries. *Adv Funct Mater* **28**, 1805899 (2018).
8. Chen L, Zhou J, Zhang J, Qi G, Wang B, Cheng J. Copper Indium Sulfide Enables Li-CO₂ Batteries with Boosted Reaction Kinetics and Cycling Stability. *ENERGY & ENVIRONMENTAL MATERIALS* **0**, 1-9 (2022).
9. Pipes R, He J, Bhargava A, Manthiram A. Efficient Li-CO₂ Batteries with Molybdenum Disulfide Nanosheets on Carbon Nanotubes as a Catalyst. *ACS Appl Energy Mater* **2**, 8685-8694 (2019).
10. Ahmadiparidari A, *et al.* A Long-Cycle-Life Lithium-CO₂ Battery with Carbon Neutrality. *Adv Mater* **31**, 1902518 (2019).
11. Chen B, *et al.* Designing Electrophilic and Nucleophilic Dual Centers in the ReS₂ Plane toward Efficient Bifunctional Catalysts for Li-CO₂ Batteries. *J Am Chem Soc* **144**, 3106-3116 (2022).
12. Qiao Y, *et al.* Li-CO₂ Electrochemistry: A New Strategy for CO₂ Fixation and Energy Storage. *Joule* **1**, 359-370 (2017).
13. Freunberger SA, Chen Y, Drewett NE, Hardwick LJ, Bardé F, Bruce PG. The Lithium-Oxygen Battery with Ether-Based Electrolytes. *Angew Chem Int Ed* **50**, 8609-8613 (2011).

Reviewer #1 (Remarks to the Author):

The answer to the question is already quite comprehensive and does not require further modification.

Reviewer #2 (Remarks to the Author):

The authors have undertaken considerable efforts to address the reviewer's concerns, which has made the work much stronger, in the opinion of the reviewer. While significant progress has been made, a few lingering issues require addressing before the study is publishable, but the reviewer believes the authors are capable of addressing these sufficiently with a few additional experiments. Most significantly, characterization of the charge and discharge reactions of CoS₂ after ~10 cycles are necessary to clarify that the stabilized cycling performance is still governed mostly by the reduction of CO₂ and the subsequent oxidation of Li₂CO₃. More detailed comments can be found below.

On line 167, the authors make the claim that "Li₂CO₃ is the main discharge product" based on the Raman and XRD evidence provided in Figures 3b and S13. Unfortunately, there is no unambiguous evidence of Li₂CO₃ after discharge on Co₉S₈ from either the Raman or XRD data. Moreover, there is no unambiguous evidence of Li₂CO₃ after discharge on CoS₂ from XRD and only a very weak signal in Raman. Therefore, this statement is weak and needs to be rewritten. However, for such small discharge capacities, it is perhaps not unsurprising that the discharge products are challenging to characterize. The authors should instead point to their titration results in Table S4, which offer more convincing evidence that Li₂CO₃ (or at least something that reacts with acid to form CO₂ gas) is the dominant discharge product.

In Table S4, the authors should add the expected amount of Li₂CO₃ based on the number of electrons passed (200 uAh/cm² I believe), which gives a value of 1.87 umol for a 2e⁻/Li₂CO₃ process. As a result, the titration data suggests that ~49-78% of the discharge process goes to the formation of Li₂CO₃ (again, assuming a 2e⁻/Li₂CO₃ process). This at least supports the claim that the major product is Li₂CO₃, although again highlights that many side reactions also occur.

The SEM evidence in Figure 3 and line 169 of the text needs to be presented more cautiously. For instance, with the resolution quality of the provided images, it is fairly challenging to see the "discharge products" cited by the authors. The authors should also clarify if the electrodes have been adequately washed following discharge to rule out the possibility that these are simply salt deposits (this isn't mentioned anywhere in the methods as far as I can tell).

For the DEMS results presented in Figure 3, please label the expected flux for charge reactions (1) and (2) on the figure based on the applied current density. The reviewer's understanding is this should be 0.155 nmol/s and 0.104 nmol/s, respectively, based on a 1 cm² electrode area and 20 uA/cm². This comparison between the theoretical and achieved CO₂ flux highlights the severe irreversibility of CoS_{1.097} and Co₉S₈ (which is in line with the titration results in Table S4).

The final concern that the review has is how the charge/discharge reaction might change with cycling. All characterizations of the charge/discharge reactions are done after the first cycle, but one of the key claims of the paper is that the structural evolution of CoS₂ to have partial oxygen substitution is key to enable its catalytic performance. As a result, it is important to also characterize the charge/discharge reaction after this structural evolution of CoS₂ has occurred. While ex-situ characterizations + titrations before and after a later cycle (for example, after the 10th discharge and 10th charge) are certainly possible, perhaps the easiest and most convincing data to address this would be DEMS data showing CO₂ evolution during charging after sufficient cycling. For the discharge reaction, pressure tracking data to show the consumption of CO₂ with cycling would also be sufficient.

Reviewer #3 (Remarks to the Author):

The revised manuscript satisfies all comments of the reviewer, thus, I recommend the acceptance

of this manuscript for publication.

Response to Reviewer #1

Comments:

The answer to the question is already quite comprehensive and does not require further modification.

Response. Thank you very much for the positive recommendations.

Response to Reviewer #2

Comments:

The authors have undertaken considerable efforts to address the reviewer's concerns, which has made the work much stronger, in the opinion of the reviewer. While significant progress has been made, a few lingering issues require addressing before the study is publishable, but the reviewer believes the authors are capable of addressing these sufficiently with a few additional experiments. Most significantly, characterization of the charge and discharge reactions of CoS_2 after ~10 cycles are necessary to clarify that the stabilized cycling performance is still governed mostly by the reduction of CO_2 and the subsequent oxidation of Li_2CO_3 . More detailed comments can be found below.

Response. Thank you very much for the positive recommendations.

On line 167, the authors make the claim that " Li_2CO_3 is the main discharge product" based on the Raman and XRD evidence provided in Figures 3b and S13. Unfortunately, there is no unambiguous evidence of Li_2CO_3 after discharge on Co_9S_8 from either the Raman or XRD data. Moreover, there is no unambiguous evidence of Li_2CO_3 after discharge on CoS_2 from XRD and only a very weak signal in Raman. Therefore, this statement is weak and needs to be rewritten. However, for such small discharge capacities, it is perhaps not unsurprising that the discharge products are challenging to characterize. The authors should instead point to their titration results in Table S4, which offer more convincing evidence that Li_2CO_3 (or at least something that reacts

with acid to form CO₂ gas) is the dominant discharge product.

Response. Thank you for the suggestion. We have rewritten the discussion about discharge products as:

“To gain mechanistic insight into the electrocatalytic process, ex situ characterizations of products on CoS_x electrodes after discharge and charge were first performed, as shown in Figure S13, including SEM (Figure S14), XRD (Figures 3a and b), Raman spectroscopy (Figure S15). The SEM images show that the discharge products are large and rodlike covering the surface of CoS_{1.097}, while those on CoS₂ and Co₉S₈ are smaller in Figures S14a-c. The XRD patterns in Figure 3a show the signal of discharge products can be assigned to Li₂CO₃ (#PDF22-1141) on CoS_{1.097}. After the charge, even though no other peaks are on all cathodes in Figure 3b, the irregular residues can be easily observed on CoS_{1.097} and Co₉S₈ while those on CoS₂ are not observable in Figures S14d-f. These results roughly indicate that, in comparison with CoS_{1.097} and Co₉S₈, CoS₂ has a higher reversibility. As the discharge products on CoS₂ and Co₉S₈ can not be clearly identified, we infer Li₂CO₃ most probably is the discharge product for the three sulfides based on previous reports and XRD pattern of discharged CoS_{1.097} in Li-CO₂ batteries. We also performed Raman spectroscopy in Figure S15 but peaks at 1080 cm⁻¹ corresponding to vibration of CO₃²⁻ in Li₂CO₃ are weak on discharged CoS₂ and Co₉S₈. To verify our assumption and quantify the reversibility for the three catalysts in Li-CO₂ batteries, titration experiments by phosphoric acid are performed on the catalysts after discharge and charge under a current density of 20 μA cm⁻² with a limited capacity of 100 μA h cm⁻², which consistent with electrochemical test (Figure S16). As shown in Figure S17, CO₂ generation after titrating acid solution on the discharged catalyst, suggesting carbonates, most likely Li₂CO₃ based on the above results, are discharge products on the three catalysts.”

In Table S4, the authors should add the expected amount of Li₂CO₃ based on the number of electrons passed (200 uAh/cm² I believe), which gives a value of 1.87 umol for a 2e⁻/Li₂CO₃ process. As a result, the titration data suggests that ~49-78% of the discharge process goes to the formation of Li₂CO₃ (again, assuming a 2e⁻/Li₂CO₃ process). This

at least supports the claim that the major product is Li_2CO_3 , although again highlights that many side reactions also occur.

Response. Thank you for the suggestion. We have added the expected amount of Li_2CO_3 and its calculation process in Table S4 in the revised SI, as shown in Table R1. Based on these results, we use Figure R1 to demonstrate the amount of Li_2CO_3 formation and residues on three cathodes and the difference between them and theoretical value. This figure has been added as Figure 3c and some sentences have been added to explain discharge reaction is a $2e^-/\text{Li}_2\text{CO}_3$ process in the revised manuscript as:

“With external standard 1# in Figure S18 and Table S3, the quantities of formed and residual Li_2CO_3 during discharge and charge on the three cathodes are shown in Figure 3c and Table S4. By now, the reported possible discharge reactions in Li- CO_2 batteries are shown as following reactions (1)-(4). The charge to mass of Li_2CO_3 in all reactions is $2e^-/\text{Li}_2\text{CO}_3$, including reaction (2) if $\text{Li}_2\text{C}_2\text{O}_4$ disproportionates to Li_2CO_3 . For a $2e^-/\text{Li}_2\text{CO}_3$ process, ~57-89% of the discharge process goes to the formation of Li_2CO_3 , indicating Li_2CO_3 -related reactions are dominant during discharge for the three sulfides.

”

Table R1. The quantities of Li_2CO_3 after discharge and charge on the three cathodes.

Cathode	Measurement value of CO_2 evolution (μmol)		Amount of Li_2CO_3 (μmol)		Discharge efficiency of Li_2CO_3 formation	Conversion efficiency of Li_2CO_3
	After discharge	After charge	After discharge	After charge		
Co_9S_8	1.46	1.18	1.68	1.37	89.8%	18.4%
$\text{CoS}_{1.097}$	0.91	0.55	1.07	0.67	57.2%	37.4%
CoS_2	1.24	0.61	1.44	0.73	77.0%	48.8%

Discharge efficiency of Li_2CO_3 formation is defined as $N_{\text{ad}}/N_{\text{th}}$.

The conversion efficiency of Li_2CO_3 is defined as $(N_{\text{ad}}-N_{\text{ac}})/N_{\text{ad}}$.

N_{ad} is the amount of Li_2CO_3 after discharge, N_{ac} is the amount of Li_2CO_3 after charge, and N_{th} is the theoretical amount of Li_2CO_3 formation after discharge. For the quantitative experiments including in situ DEMS test and titration, the discharge and charge capacity is $100 \mu\text{A h cm}^{-2}$, consistent with the electrochemical test. So N_{th} in Table S4 is $1.87 \mu\text{mol}$ for a $2e^-/\text{Li}_2\text{CO}_3$ process.

Figure R1. The amount of Li_2CO_3 formation and residues on three catalysts after discharge and charge. The dashed line is the theoretical value of Li_2CO_3 formation after discharge.

The SEM evidence in Figure 3 and line 169 of the text needs to be presented more cautiously. For instance, with the resolution quality of the provided images, it is fairly challenging to see the “discharge products” cited by the authors. The authors should also clarify if the electrodes have been adequately washed following discharge to rule out the possibility that these are simply salt deposits (this isn’t mentioned anywhere in the methods as far as I can tell).

Response. Thank you for the suggestion. We changed the SEM images with a larger scale than previously provided, as shown in Figure R2. As the pristine morphologies for all the sulfides are thin flakes vertically on carbon fibers as shown in Figure S3,

varisized particles on the cathodes are considered as reaction products after discharge and charge. The arrows are added to point out the products formed in batteries. The cathodes after discharge and charge for SEM were extracted from coin cells, washed by TEGDME in the glovebox, and dried overnight in a vacuum at 60 °C before the test. We have added this figure as Figure S14 and added the detailed method in the Characterization part in the revised SI.

Figure R2. SEM images of (a,d) Co₉S₈, (b,e) CoS_{1.097}, and (c,f) CoS₂ after discharge(upper) and recharge (down) with a limited capacity of 200 μA h cm⁻².

For the DEMS results presented in Figure 3, please label the expected flux for charge reactions (1) and (2) on the figure based on the applied current density. The reviewer's understanding is this should be 0.155 nmol/s and 0.104 nmol/s, respectively, based on a 1 cm² electrode area and 20 uA/cm². This comparison between the theoretical and achieved CO₂ flux highlights the severe irreversibility of CoS_{1.097} and Co₉S₈ (which is in line with the titration results in Table S4).

Response. Thank you for the suggestion. As the editor suggested, we changed x axis dimensions to hour and the expected flux for charge reactions are 0.559 and 0.373 μmol h⁻¹. We have labeled these as dash lines as shown in Figure R3 and updated Figure 3 in the revised manuscript.

Figure R3. DEMS results of (a) CoS₂, (b) CoS_{1.097}, and (c) Co₉S₈ during charge at a current density of 20 μA cm⁻² with a limited capacity of 100 μA h cm⁻².

The final concern that the review is has is how the charge/discharge reaction might change with cycling. All characterizations of the charge/discharge reactions are done after the first cycle, but one of the key claims of the paper is that the structural evolution of CoS₂ to have partial oxygen substitution is key to enable its catalytic performance. As a result, it is important to also characterize the charge/discharge reaction after this structural evolution of CoS₂ has occurred. While ex-situ characterizations + titrations before and after a later cycle (for example, after the 10th discharge and 10th charge) are certainly possible, perhaps the easiest and most convincing data to address this would be DEMS data showing CO₂ evolution during charging after sufficient cycling. For the discharge reaction, pressure tracking data to show the consumption of CO₂ with cycling would also be sufficient.

Response. Thank you for the suggestion. We use constant flow in the DEMS test as described in SI and thus the total pressure is maintained stable except for changing the outer flow way. The pressure data for discharging is presented in Figure R4, which is not related to discharge reactions in the battery. Therefore, we titrated the cathode after the 9th charge, 10th discharge, and 10th charge to characterize the discharge and charge reaction in the 10th cycle. As shown in Figure R5, little CO₂ generation after titrating the cathodes after the 9th and 10th charge, while an obvious CO₂ generation peak can be observed on the cathode after the 10th discharge, suggesting most Li₂CO₃ can be decomposed after charge during cycling. Since the measurement values of CO₂ generation on cathodes after the 9th and 10th charge are less than 5% of that on the

cathode after the 10th discharge, we approximate the amount of Li₂CO₃ on the cathode after the 10th discharge as the quantities of Li₂CO₃ formation and decomposition in the 10th cycle. A new external standard is used to quantify the amount of Li₂CO₃ since the DEMS equipment has been used for other experiments with different solution systems for some time past. Figures R6a-c show the titrating results for Li₂CO₃ solution with a certain concentration. The corresponding linear relationship between measurement and theoretical results is a function of $y=0.6216*x-0.13386$ in Figure R6d and Table R2, of which y is the measurement value of CO₂ evolution and x is the theoretical amount of Li₂CO₃. Based on this, the amount of Li₂CO₃ formation is ~1.15 μmol, and ~60% charge goes to form Li₂CO₃ during discharge in the 10th cycle. The above results demonstrate that Li₂CO₃ is still the main discharge product and can be almost completely decomposed during charge on the reconstructed CoS₂ in cycling.

We have added Figures R5 and 6 as Figures S20 and S21 and updated Table S3 in the revised SI. The discussion also has been added on Pages9-10 in the revised manuscript, as:

“We further titrated cathodes.... on the reconstructed CoS₂ in cycling.”

Figure R4. The pressure data from the DEMS test on CoS₂ during charge.

Figure R5. The CO₂ (m/z=44) generation after titrating CoS₂ cathodes after the (a) 9th charge, (b) 10th discharge, and (c) 10th charge. The current density of discharge and charge is 20 μA cm⁻² with a limited capacity of 100 μA h cm⁻².

The measurement values of CO₂ are 0.023, 0.58, and 0.027 μmol for CoS₂ cathodes after the 9th charge, 10th discharge, and 10th charge.

Figure R6. The CO₂ (m/z=44) generation after titrating (a) 0.1 mL, (b) 0.05 mL, and (c) 0.025 mL Li₂CO₃ solution with a certain concentration of 5.25 mg mL⁻¹. (d) The relationship between measurement value of CO₂ and theoretical amount of Li₂CO₃.

Table R2. The measurement value and theoretical amount of CO₂ evolution.

Amount of Li ₂ CO ₃ (μg)	Theoretical amount of Li ₂ CO ₃ (μmol)	Measurement value of CO ₂ evolution (μmol)
525	7.09	4.28
262.5	3.55	2.04
131.25	1.77	0.98

Response to Reviewer #3

Comments:

The revised manuscript satisfies all comments of the reviewer, thus, I recommend the acceptance of this manuscript for publication.

Response. Thank you very much for the positive recommendations

Reviewer #2 (Remarks to the Author):

The authors have diligently added all of the requested details and I have no further concerns.